# Hazards of decreasing marine oxygen: the near-term and millennial-scale benefits of meeting the Paris climate targets

Gianna Battaglia[1,2] and Fortunat Joos[1,2]

[1]Climate and Environmental Physics, Physics Institute, University of Bern, Bern, Switzerland
[2]Oeschger Centre for Climate Change Research, University of Bern, Bern, Switzerland

*Correspondence to:* Gianna Battaglia (battaglia@climate.unibe.ch)

**Abstract.** Ocean deoxygenation is recognized as key ecosystem stressor of the future ocean and associated climate-related ocean risks are relevant for policy decisions today. In particular, benefits of reaching the ambitious 1.5 °C warming target mentioned by the Paris Agreement compared to higher temperature targets are of high interest. Here, we model oceanic oxygen, warming, and their compound hazard in terms of metabolic conditions on multi-millennial timescales for a range of equilibrium
temperature targets. Scenarios, where radiative forcing is stabilized by 2300, are used in ensemble simulations with the Bern3D Earth System Model of Intermediate Complexity. Transiently, the global mean ocean oxygen concentration decreases by a few percent under low and by 40 % under high forcing. Deoxygenation peaks about thousand years after stabilization of radiative forcing and new steady state conditions establish after AD 8000 in our model. Hypoxic waters expand over the next millennium and recovery is slow and remains incomplete under high forcing. Largest transient decreases in oxygen
are projected for the deep sea. Distinct and close to linear relationships between the equilibrium temperature response and marine $O_2$ loss emerge. These point to the effectiveness of the Paris climate target in reducing marine hazards and risks. Mitigation measures are projected to reduce peak decreases in oceanic oxygen inventory by 4.4 % °C$^{-1}$ of avoided equilibrium warming. In the upper ocean, the decline of a metabolic index, quantified by the ratio of $O_2$ supply to an organism's $O_2$ demand, is reduced by 6.2 % °C$^{-1}$ of avoided equilibrium warming. Measures of peak hypoxia exhibit a strong sensitivity to
additional warming. Volumes of water with less than 50 mmol $O_2$ m$^{-3}$, for instance, increase between 36 % to 76 % °C$^{-1}$ of equilibrium temperature response. Our results show that millennial-scale responses should be considered in assessments of ocean deoxygenation and associated climate-related ocean risks. Peak hazards occur long after stabilization of radiative forcing and new steady state conditions establish after AD 8000.

# 1 Introduction

Oxygen ($O_2$) is a sparingly soluble gas and its abundance in the ocean is decreasing under ongoing global warming (*IPCC*, 2013). Decreasing $O_2$ concentrations, warming and changes in other environmental parameters forced by anthropogenic greenhouse gas (GHG) emissions pose high risks for marine ecosystems (*Gattuso et al.*, 2015). The parties of the United Nations Framework Convention on Climate Change (*UNFCCC*, accessed 6. February 2018) and of the Paris Agreement note 'the importance of ensuring the integrity of all ecosystems, including Oceans' (*Paris Agreement*, accessed 10. April 2018) and 'the threats of irreversible damage' (*UNFCCC*, accessed 6. February 2018, Article 3). Marine changes are projected to evolve on multi-century and millennial time scales with peak impacts occurring potentially long after stabilization of atmospheric GHG concentrations and peak temperatures. Yet, only few studies assess millennial scale impacts of anthropogenic GHG emissions on the ocean and the reversibility of marine changes in oxygen. Explicit quantification of the benefits of meeting the 2°C or 1.5°C climate targets mentioned by the Paris Agreement with respect to the reversibility and avoidance of implied impacts on marine oxygen and related environmental parameters, including ocean circulation, ocean warming, metabolic viability and biological productivity on multi-millennial timescales is missing.

Typical thresholds leading to $O_2$-stress for many macroorganisms (hypoxia) are around 50 mmol $O_2$ m$^{-3}$. Water with lower $O_2$ concentrations are effectively dead zones for many higher animals (reviewed in *Keeling et al.*, 2010; *Storch et al.*, 2014; *Breitburg et al.*, 2018). Species are also sensitive to thermal stress (*Gattuso et al.*, 2015) and their sensitivity to hypoxia increases with higher temperatures (*Pörtner*, 2010). In the modern ocean, oxygen-poor zones with $O_2 < 50$ mmol m$^{-3}$ occupy about 5 % of its volume (*Garcia et al.*, 2014; *Bianchi et al.*, 2012). Expanding oxygen-poor waters lead to habitat compression, mortality and major changes in community structure where energy preferentially flows into microbial pathways to the detriment of higher trophic levels. Suboxic (<5 mmol $O_2$ m$^{-3}$) or anaerobic conditions can also lead to production of poisonous $H_2S$ within sediments (reviewed in *Diaz and Rosenberg*, 2008; *Breitburg et al.*, 2018) and decreasing $O_2$ concentrations potentially lead to higher production and emissions of the greenhouse gas nitrous oxide.

Observational (*Schmidtko et al.*, 2017; *Ito et al.*, 2017) and modeling studies (*Oschlies et al.*, 2017) indicate an overall decline in the oceanic oxygen content over past decades. Systematic discrepancies exist for the typically low oxygen tropical thermocline, where observations suggest $O_2$ has decreased and most models simulate increased $O_2$ levels over the past decades. Model projections to the end of the 21st century consistently project the global ocean oxygen inventory to further decline with anthropogenic climate change (*Matear*, 2000; *Plattner et al.*, 2001; *Bopp et al.*, 2002; *Frölicher et al.*, 2009; *Cocco et al.*, 2013; *Bopp et al.*, 2013). The most recent generation of Earth system models simulate global deoxygenation by the end of the 21st century of round -1.81% (RCP2.6) to -3.45% (RCP8.5) (*IPCC*, 2013). Impact studies have highlighted potential habitat compression (*Deutsch et al.*, 2015; *Mislan et al.*, 2017) and reduced catch potential (*Cheung et al.*, 2016) associated with climate change at the end of the century. Large model-model differences remain in projections of oxygen minimum zones (OMZs) (*Cocco et al.*, 2013; *Bopp et al.*, 2013).

Given the long residence time of anthropogenic $CO_2$ in the atmosphere, and long equilibration timescales of the ocean overturning circulation, anthropogenic climate change will grow and persist beyond the end of the 21st century, the typical near-term assessment timescale of climate change (*Clark et al.*, 2016). Only few studies have assessed ocean biogeochemistry and the oceanic oxygen content beyond this near-term timescale. Available studies employ a range of physical and biogeo-chemical complexity levels from box models to general circulation models (GCMs). Oxygen concentrations are simulated to decline beyond the 21st century on multi-centennial timescales (*Matear and Hirst*, 2003; *Hofmann and Schellnhuber*, 2009; *Mathesius et al.*, 2015). Simulations covering two millennia show a recovery phase thereafter (*Schmittner et al.*, 2008; *Yamamoto et al.*, 2015). In most studies, simulated oxygen concentrations have not reached new steady state conditions at the end of the simulation. Low order Earth system models and Earth System Models of Intermediate complexity integrated by up to 100,000 years have demonstrated the potential for long-term ocean oxygen depletion in response to carbon dioxide emissions and the long equilibration time scales of ocean biogeochemical variables in response to carbon emissions (*Shaffer et al.*, 2009; *Ridgwell and Schmidt*, 2010). Multi-millennial simulations are therefore required to assess the full amplitude of ocean biogeochemical changes and new steady state conditions due to anthropogenic climate change.

The distribution of $O_2$ in the ocean results from the sum of its solubility component set trough air-sea exchange, the effect of $O_2$ production by phytoplankton in the euphotic zone and $O_2$ consumption during organic matter remineralization at depth. In modeling studies, it is possible to identify the drivers of $O_2$ changes by considering changes due to solubility and changes due to oxygen consumption. When assessing the near-term timescale at the end of 21st century, studies have shown that different depths in the water column tend to be associated with different dominant mechanisms of change. In the surface ocean, $O_2$ changes tend to be determined by changes in $O_2$ solubility. In the subsurface, both changes in solubility and utilization may reinforce (mid and high latitudes) or compensate each other (tropics) (e.g. *Cabre et al.*, 2015; *Bopp et al.*, 2017). In the deep ocean, simulated $O_2$ changes are dominated by changes in $O_2$ utilization, which is in turn controlled by ocean ventilation (see also *Matear and Hirst*, 2003; *Yamamoto et al.*, 2015, for longer timescales). Changes in the oceanic heat content and in ocean circulation are therefore crucial for $O_2$ changes.

Well-defined metrics that summarize the Earth system response are useful in many aspects and may facilitate the communica-tion in the mitigation policy context of the Paris agreement. The Transient Climate Response to Cumulative Carbon Emissions (TCRE, *Allen et al.*, 2009) or the Transient Earth System Response to Cumulative Carbon Emissions (TREX, *Steinacher and Joos*, 2016) are such metrics. These link changes in global surface air temperature and environmental parameters to cumulative $CO_2$ emissions relying on near linear relationships. Similar metrics may be developed for oceanic oxygen.

Deoxygenation is one of several marine ecosystem stressors including warming, acidification, hypocapnia, changes in food supply and sea-level rise (*Gruber*, 2011; *Cocco et al.*, 2013; *Bopp et al.*, 2013; *Gattuso et al.*, 2015; *Sweetman et al.*, 2017). Several key marine and coastal ecosystems may face high risk of impact due to climate change even if low emission pathways are followed to the end of the century (*Gattuso et al.*, 2015; *Magnan et al.*, 2016; *Breitburg et al.*, 2018). This growing body of concern contributed to motivate the 21st Conference of the Parties (COP21) to reach the Paris Agreement. A goal of the

Paris Agreement is to 'hold the increase in the global average temperature to well below 2°C above pre-industrial levels and pursuing efforts to limit the temperature increase to 1.5°C above pre-industrial levels' (Article 2a, *Paris Agreement*, accessed 10. April 2018). Article 4 of the Paris Agreement further states that 'In order to achieve the long-term temperature goal set out in Article 2, Parties aim to reach global peaking of greenhouse gas emissions as soon as possible .. and to undertake rapid reductions thereafter in accordance with best available science, so as to achieve a balance between anthropogenic emissions by sources and removals by sinks of greenhouse gases in the second half of this century (Article 4, *Paris Agreement*, accessed 10. April 2018).

In this study, we assess the effectiveness of the Paris climate targets in reducing hazards of decreasing oceanic oxygen, ocean warming and marine export productivity as simulated by the Bern3D Earth system model of intermediate complexity. We prescribe in the model four different, idealized scenarios where anthropogenic GHG forcing is stabilized by 2300 AD. The four scenarios are designed to reach an equilibrium warming of 1.5, 1.9, 3.3 and 9.2°C above preindustrial. Simulations are run to year AD 10,000 by which time the ocean has reached new steady state conditions. This allows us to assess reversibility and the full amplitude of changes, acknowledging the long equilibration timescale of biogeochemical variables with peak hazards potentially occurring long after stabilization of radiative forcing in the atmosphere. We summarize the outcomes developing global metrics which quantify avoided marine hazards per avoided global warming on three different time horizons. The first time horizon is the end of the 21st century, the typical assessment timescale of climate change hazards. Here, changes in a variable are related to changes in SAT at year 2100. Those are contrasted to the millennial-scale perspective where peak changes in the variable in the course of the simulation and equilibrium changes at the end of the simulation are related to the corresponding equilibrium warming.

In section 2, we briefly describe the Bern3D model and the experimental setup. Four different radiative forcing stabilization scenarios to meet four temperature targets (1.5, 1.9, 3.3 and 9.2°C above preindustrial) are considered. The observation-constrained 100-member ensembles used to explore parameter uncertainties for each scenario is introduced. In section 3, physical changes, including changes in overturning, water mass age, sea ice, temperature, salinity and density as well as biogeochemical changes, including changes in global oxygen inventory, the extent of oxygen minimum zones, and productivity are presented. The compound effects of warming and oxygen changes are assessed in the form of a metabolic index (*Deutsch et al.*, 2015). Underlying physical and biogeochemical processes and mechanisms are discussed. Following earlier studies, we attribute the contributions of $O_2$ changes from changes in solubility, and the interplay of ocean biology and ventilation by carrying four explicit $O_2$ tracers and an ideal age tracer. The graphical illustration of spatial changes is focused on a 1.5°C equilibrium warming target at the point of peak $O_2$ decline. Additional supporting figures are given in the appendix. In section 4, the relationship between change in global mean surface air temperature (ΔSAT) and selected impact-relevant parameters is quantified. The different relationships are established for the near-term (2100 AD), the time of the peak decline in oxygen around 3000 to 4000 AD, and at year 10,000 AD when a new equilibrium has been reached in the model. Often a near-linear relationship is found between the change in a variable of interest and the change in SAT as simulated across

the range of scenarios and ensemble members at a distinct time. This allows us to develop new metrics to quantify avoided marine hazards per unit change in $\Delta$SAT at different points in time. These quantitatively illustrate the benefits of meeting the Paris target in terms of marine hazards. Each modeling exercise is associated with uncertainties and in section 5, we discuss relevant uncertainties, mention neglected processes and compare our findings to other studies. Finally, in section 6 we present

implications and conclusions and summarize our findings graphically for a '1.5°C world' and contrast peak changes across the range of temperature targets.

## 2   Model and Simulations

### 2.1   Bern3D

The Bern3D Earth System Model of Intermediate Complexity is a three dimensional frictional geostrophic balance ocean

model (*Müller et al.*, 2006), which includes a sea ice component coupled to a single-layer energy and moisture balance model of the atmosphere (*Ritz et al.*, 2011) and a prognostic marine biogeochemistry module (*Tschumi et al.*, 2011; *Parekh et al.*, 2008). A version with 41x40 horizontal grid-cells and 32 vertical layers is used (see also *Roth et al.*, 2014; *Battaglia et al.*, 2016 for model evaluation). The NCEP/NCAR monthly wind-stress climatology (*Kalnay et al.*, 1996) is prescribed at the surface. Air-sea gas exchange, carbonate chemistry and natural $\Delta^{14}$C of DIC is modeled according to OCMIP-2 protocols

(*Najjar et al.*, 1999; *Orr and Najjar*, 1999; *Orr and Epitalon*, 2015). The global mean air-sea transfer rate is reduced by 19 % compared to OCMIP-2 to match observation-based estimates of natural and bomb-produced radiocarbon (*Müller et al.*, 2008).

     The biogeochemical module is based on phosphorus and simulates production and remineralization/dissolution of organic matter, calcium carbonate and opal. Production of particulate organic matter (POP) within the euphotic zone (top 75 m) depends on temperature, light availability, phosphate and iron following *Doney et al.* (2006). POP remineralization within the water

column follows a power law profile (*Martin et al.*, 1987). Organic matter falling on to the sea floor is remineralized in the deepest box. Two thirds of organic matter production form dissolved organic matter (DOP), which decays with an e-folding lifetime of 1.5 years. An updated remineralization scheme assigns remineralization of POP and DOP to aerobic and anaerobic pathways depending on the mean grid-cell dissolved $O_2$ concentration (see *Battaglia and Joos* (2018)). We introduce two power law profiles with two distinct remineralization length scales for aerobic and anaerobic remineralization ($\alpha_{\mathrm{aerob}}$ and

$\alpha_{\mathrm{denit}}$). Constant stoichiometric ratios are used for both aerobic and anaerobic remineralization to convert biological P fluxes into carbon, and alkalinity fluxes (P:Alk:C=1:17:117). The $O_2$ demand for complete aerobic remineralization is 170 $\frac{molO_2}{molPO_4}$ and no oxygen is consumed for anaerobic remineralization. Accordingly, aerobic remineralization in the ocean is smaller than $O_2$ production in the euphotic zone leading to an $O_2$ outgassing for steady state conditions. The atmospheric oxygen inventory is constant. This is justified as 99.5 % of the ocean-atmosphere inventory is in the atmosphere and potential net fluxes of $O_2$

from the ocean and land to the atmosphere and fossil fuel burning have a small impact on atmospheric $O_2$. $O_2$ components

from $O_2$ production, consumption and solubility are carried as explicit model tracers to attribute changes. Tracers add up to within $10^{-14}$ Pmol with mean inventories of 23.2, -239, 430 yielding a total of 214 Pmol, respectively (median values given). $O_2$ components inferred from $O_2$ saturation can result in systematic errors from surface disequilibrium (*Ito et al.*, 2004). The use of explicit tracers avoids such systematic errors in the $O_2$ components. As changes in the $O_2$ production term are small, we

combine the $O_2$ production and consumption tracers to a $O_2$ biology tracer when displaying sections.

We include evaluation of a metabolic index, $\Phi$, which was proposed by *Deutsch et al.* (2015). It combines temperature and $pO_2$ as indicators of metabolically viable environments and is defined as the ratio of $O_2$ supply to an organism's resting $O_2$ demand. We consider only relative changes in $\Phi(t)$ relative to a reference time, $t_0$ (average over 1870-1899):.

$$\frac{\Delta\Phi(t)}{\Phi(t_0)} = \frac{pO_2(t)}{pO_2(t_0)} \times exp\left(\frac{E_0}{k_B}\left(\frac{1}{T(t)} - \frac{1}{T(t_0)}\right)\right) - 1, \tag{1}$$

where $T$ is the absolute temperature, $k_B$ is Boltzmann's constant and the exponential function and the parameter $E_0$ characterize the temperature dependence of the baseline metabolic rate. $E_0$ only weakly affects the relative influence of temperature and $O_2$ gradients and relative changes in $\Phi$ are therefore independent on species (*Deutsch et al.*, 2015). Here, we consider $E_0$=0.87 eV (representative of Atlantic cod). For the calculation of $pO_2$ we pressure-correct the equilibrium constant following Eq. 5 in *Weiss* (1974). The metabolic index $\Phi$, as proposed by *Deutsch et al.* (2015), is linear in $pO_2$ (representing the rate

of $O_2$ supply) and decreases non-linearly with temperature (indicative of the resting metabolic demand). One may note that the exponential curve varies approximately linearly for typical global warming associated temperature changes as $E_0/k_b (\approx 10,000$ K) is large.

The current set up does not include sediment interactions, temperature dependent remineralization, variable stoichiometry, nitrogen-cycle feedbacks, atmospheric nutrient deposition, dynamic wind nor freshwater input/albedo changes from melting

of continental ice-sheets.

## 2.2 Ensemble and scenarios

To explore potential oxygen changes we set up four 100-member ensembles each targeting a different equilibrium temperature response ($\sim$1.5, 1.9, 3.3 and 9.2 °C above preindustrial). A feedback parameter $\lambda$ [W m$^{-2}$ K$^{-1}$] (*Ritz et al.*, 2011), accounting for climate feedbacks that are not explicitly treated in the Bern3D model, is chosen in combination with radiative forcing from

the Representative Concentration Pathways (RCPs) (*Meinshausen et al.*, 2011) to achieve these stabilization targets. RCP2.6, stabilizing by 2300, is run with $\lambda$ values of -0.71 and -1 W m$^{-2}$ K$^{-1}$ achieving the 1.5 and 1.9 °C targets, respectively. RCP4.5, stabilizing after 2100, is run with -1 W m$^{-2}$ K$^{-1}$ yielding a 3.3 °C temperature response and RCP8.5, stabilizing in the 23rd century, with -0.71 W m$^{-2}$ K$^{-1}$ yielding a 9.2 °C response (median values given for temperature targets). Each member is

spun up over 5000 years to AD 1765 boundary conditions. The radiative forcing follows RCP scenarios (RCP2.6, 4.5 and 8.5, *Meinshausen et al.*, 2011). The RCP scenarios are extended to year AD 10,000 by which time the ocean has equilibrated to new steady state conditions. Radiative forcing includes an 11-year solar cycle up to year AD 3000. After that, all forcings are kept constant. We employ a single-model setup, and assess uncertainties arising from organic matter remineralization ($\alpha_{\mathrm{aerob}}$

and $\alpha_{\mathrm{denit}}$) and vertical mixing ($k_{diff-dia}$). The three parameters are sampled using the Latin Hypercube sampling technique (*McKay et al.*, 1979). The parameter ranges are chosen such that all members achieve similar skill scores with respect to observation-derived fields of natural radiocarbon (*Key et al.*, 2004) and dissolved $O_2$ (*Garcia et al.*, 2014; *Bianchi et al.*, 2012) and correspond to the values chosen in *Battaglia and Joos* (2018, Table 1). A normal distribution is used to sample $\alpha_{\mathrm{aerob}}$ with a standard value of -0.83 and a standard deviation of -0.0625. $\alpha_{\mathrm{denit}}$ is sampled uniformly between -0.1 and -0.01.

And a lognormal distribution is used to sample $k_{diff-dia}$ (standard value=2.25E-5 $m^2$ $s^{-1}$, shape parameter=0.2, location parameter=0). We choose a single ensemble member with parameter values close to the standard values as representative ensemble member to illustrate spatial anomalies ($\alpha_{\mathrm{aerob}}$=-0.85, $\alpha_{\mathrm{denit}}$=-0.037, $k_{diff-dia}$=2.05E-05 $m^2$ $s^{-1}$).

## 2.3 Pre-Industrial characteristics

The ensemble produces a range in overturning strengths, remineralization fluxes and $O_2$ distributions. The following numbers

represent the 90 % confidence ranges of important model characteristics across the ensemble. The maximum of the Atlantic meridional overturning streamfunction below 400 m depth (AMOC) ranges from 16.5 to 19.7 Sv. The minimum of the Indo-Pacific meridional overturning streamfunction below 400 m depth (Indo-Pacific MOC) ranges between -13.6 to -15.6 Sv. Export of particulate organic matter at 75 m ranges from 9.0 to 11.4 Gt C $yr^{-1}$. The simulated oxygen inventory ranges between 195 and 230 Pmol given the three parameters and the simulated oxygen distribution covers the observational range and spatial

pattern well (see Fig. 3, Fig. 7a, Table D1 of *Battaglia and Joos*, 2018). Biases exist in the simulated extend of OMZs. The volume of suboxic conditions ($O_2 < 5$ mmol $m^{-3}$) is overestimated by a factor of five but water column denitrification fluxes are well within current estimates (Table D.1., Fig. 2c in *Battaglia and Joos*, 2018). This is a common model bias in EMICs and GCMs (*Cocco et al.*, 2013; *Bopp et al.*, 2013; *Cabre et al.*, 2015). Vastly enhanced spatial resolution may be required to simulate equatorial physics and ecosystems in better agreement with observations (*Bopp et al.*, 2013).

## 3 Marine changes in temperature, circulation and biogeochemistry

We first describe the evolution of important physical quantities that impact $O_2$ concentrations. Figure 1 displays the temporal changes in global mean surface air and ocean temperature, the evolution of annual mean sea-icea area in the Northern and Southern Hemisphere, and the Atlantic and Indo-Pacific meridional overturning circulation.

In response to the RCP scenarios, atmospheric temperatures rise and stabilize after ∼1000 years (Fig. 1a). The four ensembles reaching 1.5, 1.9, 3.3 and 9.2 °C above preindustrial surface air temperature show an equilibrium ocean warming of 1.1, 1.3, 2.0 and 5.5 °C, respectively (median values given). Sea ice retreats in both hemispheres (Fig. 1c,d). The retreat is more pronounced for higher forcing. In the Southern Hemisphere, even the lower forcing levels show strong decline in the annual mean sea-ice area and sea ice vanishes for higher forcing. The warming perturbation causes the AMOC and Indo-Pacific MOC
to decline transiently (Fig. 1e,f, Fig. A1). The larger the forcing and implied changes in stratification, the larger the peak decline in overturning (Fig. 1e,f). The decline is likely driven by upper ocean warming, leading to increasing surface-to-deep density gradients as further modulated by salinity changes (Fig. A2). The deep ocean water mass age increases in response to the slowed overturning (Fig. 2d, 3d). As retreating sea-ice increases wind stress over these newly exposed areas, younger water masses form in the upper ocean of the Southern Ocean (Fig. 3d). Reduced convection may contribute to a younger upper
ocean through decresed entrainment of old deep water (not quantified within the scope of this paper). As the model tends to equilibrate under the sustained radiative forcing, the surface-to-deep gradients in the density anomalies diminish (Fig. A2), the meridional overturning circulation recovers (Fig. 1e,f), and anomalies in water mass age become again smaller (Fig. 3d versus Fig. A3). The final circulation state is close to but not identical to the preindustrial steady state circulation. Maximum overturning strength in AMOC and the Indo-Pacific MOC varies by less than $\pm$ 1 Sv around the initial value. At the new steady
state, the maximum in AMOC below 400 m tends to be lower under higher forcing, whereas the maximum in the Indo-Pacific MOC below 400 m tend to be higher under higher forcing (Fig. 1e,f). It is difficult and beyond the scope of this paper to conclusively explain such subtle changes in ocean dynamics and overturning (Fig. A1), likely linked to the complex changes in density (Fig. A2) and sea ice retreat (Fig. 1c,d). Yet, these differences have direct consequences for the projected global water mass age and by that for oceanic oxygen (Fig. 2) at the new equilibrium as further discussed below.

The response in oceanic oxygen is complex and characterized by an initial decline followed by a recovery phase (Fig. 2a). In line with earlier studies (*Matear and Hirst*, 2003; *Schmittner et al.*, 2008; *Shaffer et al.*, 2009; *Ridgwell and Schmidt*, 2010; *Mathesius et al.*, 2015), our results demonstrate the potential for large changes in marine oxygen under anthropogenic forcing, a large inertia in the response and a slow, and partially incomplete recovery of the perturbation. Transiently, the whole ocean oxygen inventory decreases by a few percent (6 %) under low forcing and by as much as 40 % under high forcing (median
values given). The minimum in oxygen occurs about thousand years after stabilization of radiative forcing, and it takes several millennia to approach a new equilibrium. Then, the global ocean $O_2$ inventory is a few percent higher than at preindustrial conditions under low and intermediate forcing and remains depleted by around 8 % in the high forcing case.

Figure 2 further explains the temporal evolution and interplay of the underlying drivers. In all cases, the changes in global oxygen inventory (Fig. 2a) strongly correlate with water mass age (Fig. 2d) and are also impacted by gradual oxygen loss due to warming as evidenced by the evolution of the $O_2$ solubility tracer (Fig. 2b). Inventory changes based on the $O_2$ production tracer (Fig. 2c) are negligible; changes equilibrate with the atmosphere and only a small fraction remains in the ocean. The $O_2$ consumption tracer (Fig. 2e) determines the shape of the global $O_2$ signal (Fig. 2a). Its decline and recovery phase is strongly correlated with the evolution of ideal age (Fig. 2e and Fig. 2d, see also Fig. 3b and Fig. 3d). As has been shown previously (e.g. *Yamamoto et al.*, 2015; *Bopp et al.*, 2017), the high correlation between changes in $O_2$ and ideal age and the absence of a direct relationship between changes in remineralization fluxes and $O_2$, indicate that circulation changes are the major contributors to changes in $O_2$. Changes from remineralization fluxes include both changes in absolute aerobic remineralization fluxes and changes in the relative share of denitrification (Fig. 2i). An increased share of denitrification at organic matter remineralization, for instance, effectively constitutes an implicit $O_2$ gain. Denitrification fluxes correlate with the volumetric expansion of OMZs and are also impacted by changes in remineralization fluxes within them (Fig. 2i). The recovery level of the $O_2$ consumption tracer (Fig. 2e) reflects the global recovery level of ideal age (Fig. 2d), where younger water masses are associated with less $O_2$ consumption and therefore higher $O_2$ concentrations. The total $O_2$ recovery level (Fig. 2a), on the other hand, is diminished due to $O_2$ loss from solubility (Fig. 2b). As such, 1.5 to 3.3 °C warming targets reach similar global $O_2$ equilibrium levels for different reasons. The 1.9 and 3.3 °C warming targets tend to result in younger water masses, which would increase $O_2$ due to less $O_2$ consumption compared to 1.5 °C warming targets. As those scenarios are also associated with higher warming, they lose more $O_2$ due to less solubility compared to 1.5 °C warming targets and yield similar global anomalies despite more pronounced spatial patterns. The 9.2 °C warming target reaches a lower equilibrium $O_2$ inventory compared to preindustrial due to high $O_2$ loss from solubility (-44.1 Pmol).

We illustrate spatial changes in critical variables for a single, representative ensemble member (see Section 2.2) at its peak $O_2$ decline which occurs at year AD 3150 and amounts to 5 % (Fig. 3). The member eventually reaches a 1.5 °C warming target. Figure 3 displays anomalies in total $O_2$ (Fig. 3a), and the contributions from biologically-mediated changes (termed "biology" below, Fig. 3b) combining the changes in the $O_2$ production and consumption tracer and from changes in solubility (Fig. 3c). In the upper ocean $O_2$ concentrations tend to increase due to biology and decrease due to solubility. Such compensating mechanisms have been documented elsewhere (e.g. *Cabre et al.*, 2015; *Bopp et al.*, 2017). The resulting changes in $O_2$ are less pronounced than the changes in each component. The increase in $O_2$ due to biology stems from younger water masses and less export in the low- and mid latitudes (see next paragraph and Fig. 4c). $O_2$ changes show strong spatial correlation with changes in water mass age (Fig. 3a,d). Largest decreases in $O_2$ are simulated in bottom waters in line with older water mass age. The equilibrium response in $O_2$ for this 1.5 °C warming case is characterized by slight $O_2$ decreases in the Atlantic, caused mainly by less solubility, and increases in the Southern Ocean and deep Pacific, caused by higher overturning and less sea-ice coverage in the Southern Ocean compared to preindustrial (Fig. A3).

Global export production is simulated to decline over the first few centuries, and reach higher values under new steady state conditions (Fig. 2g). The decline is stronger for higher forcing, while the recovery level of global export production is similar across the scenarios. Bern3D transiently simulates decreased export in the mid- and low latitudes (Fig. 4c, see also *Steinacher et al.* (2009); *Battaglia and Joos* (2018)) as a result of increased stratification (Fig. A2c,f,i) and reduced nutrient

concentrations in the surface ocean (Fig. 4b). In the high latitudes, the model simulates increased export production, as a result of less temperature and light limitation as surface waters warm and sea ice retreats. This pattern of decreased export in mid- and low latitudes and increased export in high latitudes is similar across the scenarios. Export production in the low latitudes fully recovers for lower forcing and partially recovers for higher forcing. The lower recovery level in the low latitudes is compensated by higher increases in the high latitudes for high forcing. The magnitude of positive and negative changes

increases with forcing, but the global anomalies remain comparable at the end of the simulation.

Next to changes in export, we consider the evolution of a metabolic index in the upper ocean which integrates effects of changes in $O_2$ and temperature at the organism level (Fig. 2f and Fig. 4e). The globally averaged, upper ocean (depth < 400 m) metabolic index declines throughout the simulation dominated by increased temperatures (Fig. 2f). The metabolic index, $\Phi$ (*Deutsch et al.*, 2015), decreases in most places in line with warming and lower $pO_2$ (Fig. 4a,d,e). The $O_2$ gain in upper ocean

waters is able to counteract the adverse effect of warming in some high latitude environments. In other places with higher $pO_2$, the temperature increase dominates the response in $\Phi$. Near bottom waters in the Pacific are prone to largest reductions in $\Phi$, driven by large decreases in $pO_2$ (Fig. 4e).

Oxygen-poor waters ($O_2 < 50$ mmol m$^{-3}$, Fig. 2h) are simulated to transiently increase across all scenarios. The response is characterized by high uncertainty as introduced by the sampled parameters. Under new equilibrium conditions, the volume

of low $O_2$ waters is reduced for low and intermediate forcing and remains higher than pre-industrial in the high forcing case.

Turning to uncertainties in our perturbed parameter ensemble, we find that variations in the vertical diffusion parameter ($k_{diff-dia}$) dominate the uncertainty in the globally-averaged evolution of ideal age, sea ice cover, temperature and $O_2$. The modeled uncertainty in the volume of low $O_2$ waters is dominated by different values of the $\alpha_{aerob}$ parameter. Whether a threshold in $O_2$ concentration is met depends on the pre-industrial tracer distribution. Longer remineralization length scales

bring more remineralization to depth, leading to higher $O_2$ consumption.

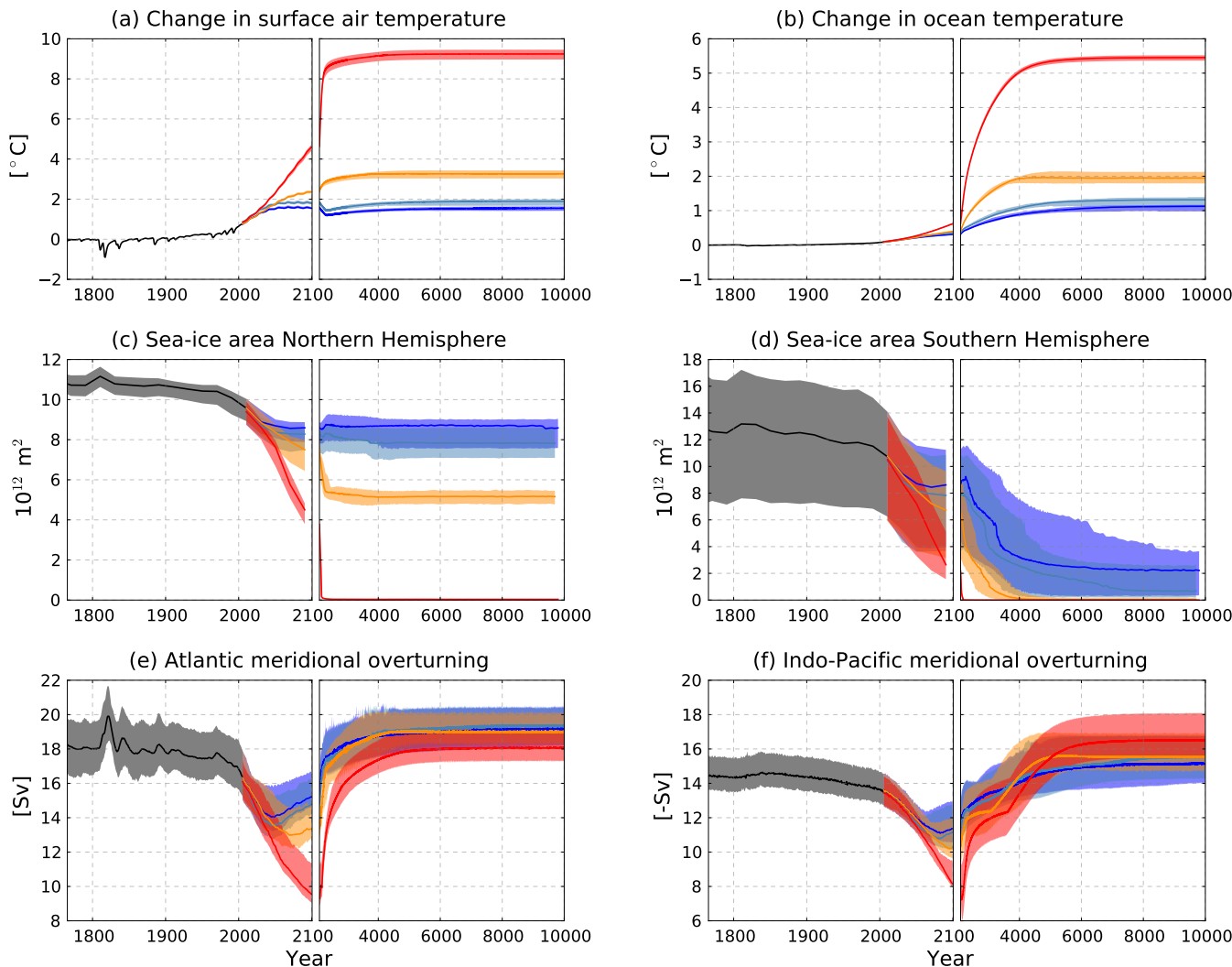

**Figure 1.** Temporal evolution of physical variables relative to 1870-1899 for model ensembles aiming at 1.5, 1.9, 3.3 and 9.2 °C warming targets. Lines mark the median and shading marks the 90 % range of the ensemble. The shading reflects uncertainties due to variations in the diapycnal mixing coefficient. c) and d) are annual mean sea-ice areas in the respective hemisphere. e) Atlantic meridional overturning is the maximum of the Atlantic and f) Indo-Pacific meridional overturning is the minimum of the Indo-Pacific meridional overturning streamfunction below 400 m depth.

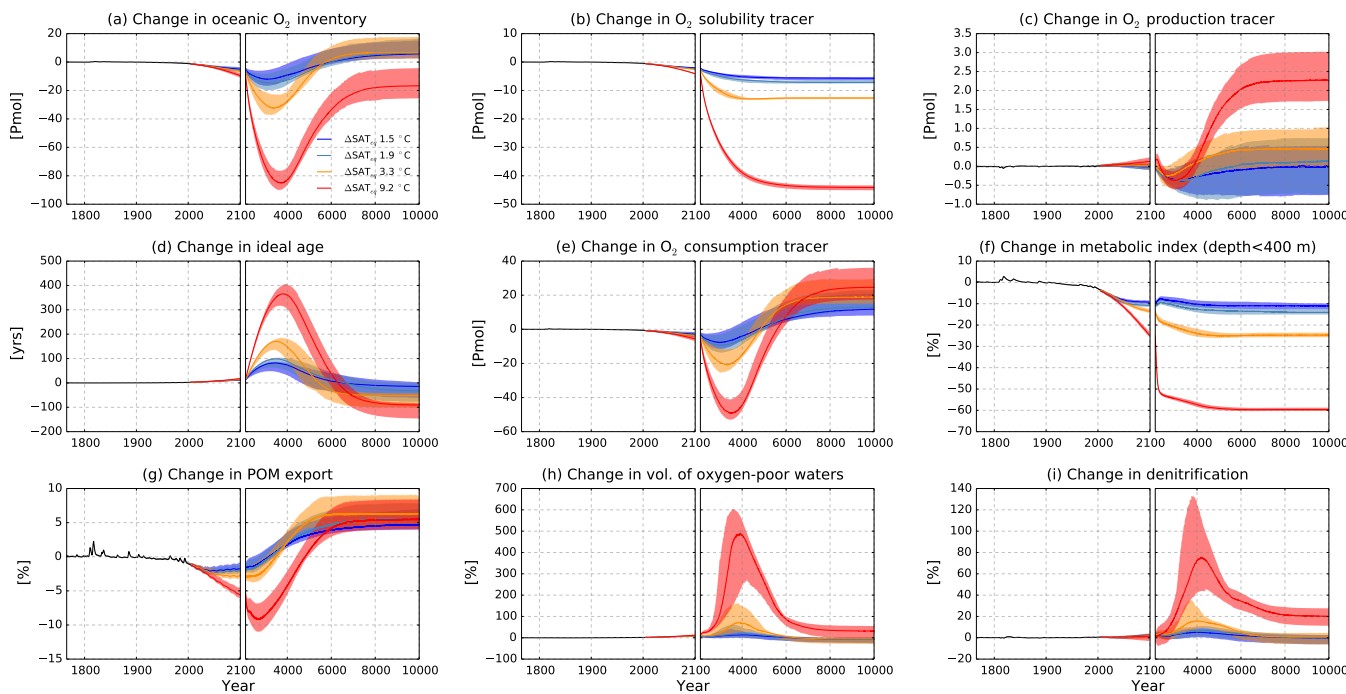

**Figure 2.** Temporal evolution of critical variables relative to 1870-1899 for model ensembles aiming at 1.5, 1.9, 3.3 and 9.2 °C warming targets. Lines mark the median and shading marks the 90 % range of the ensemble. The shading reflects uncertainties due to variations in the diapycnal mixing coefficient, and the aerobic and anaerobic remineralization length scales of particulate organic matter ($\alpha_{\mathrm{aerob}}$ and $\alpha_{\mathrm{denit}}$). b) $O_2$ solubility is the explicitly traced solubility component of oceanic oxygen, c) is the explicit $O_2$ production tracer, e) is the explicit $O_2$ consumption tracer. h) Oxygen-poor waters are taken as the volume of water with $O_2 < 50$ mmol m$^{-3}$.

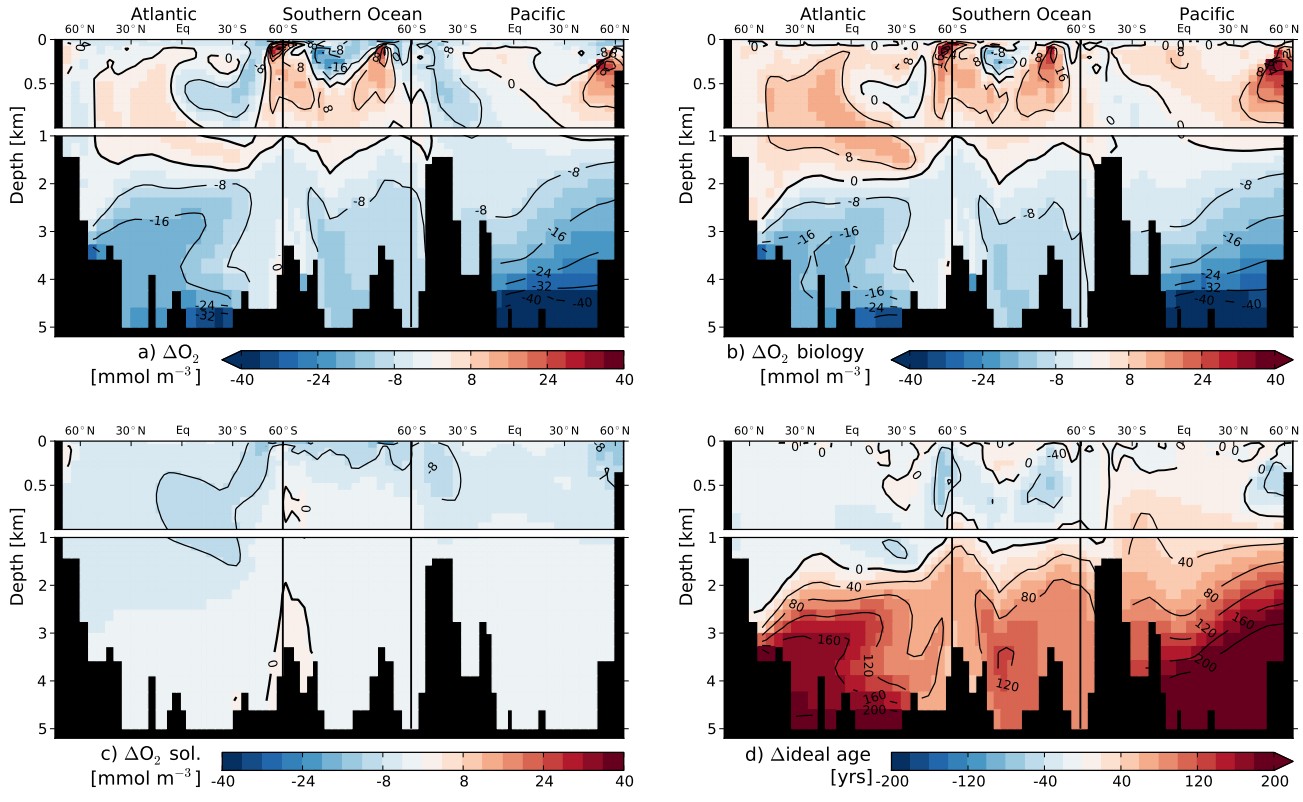

**Figure 3.** Changes in O₂ and its components at time of peak O₂ decline (AD 3150) relative to preindustrial steady state for a single, representative ensemble member reaching a 1.5 °C warming target. a) Change in total O₂, b) change in O₂ due to biology, c) change in O₂ due to solubility, d) change in ideal age.

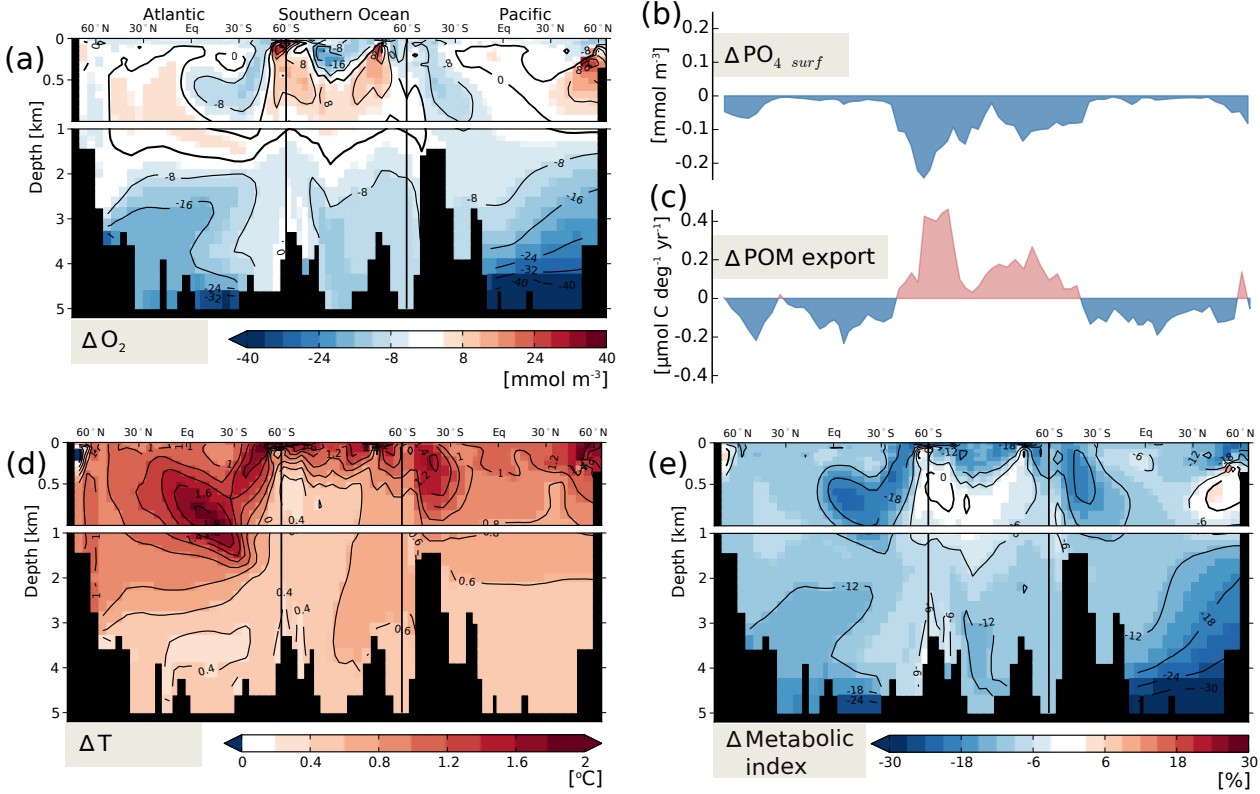

**Figure 4.** Changes in potential ecosystem stressors at peak $O_2$ decline (AD 3150) relative to preindustrial steady state for a single, representative ensemble member reaching a 1.5 °C warming target. Results are displayed for a cross section through the Atlantic (25° W), across the Southern Ocean (58° S) and into the Pacific (175° W). Changes in POM export at 75 m (c) and in surface $PO_4$ concentrations (d) are displayed along the same section.

## 4 Metrics linking global warming to marine hazards

The purpose of this section is to quantify the relationship between changes in global mean surface air temperature (SAT), the target variable of the Paris Agreement, with selected aggregated metrics for marine ecosystem stressors. In this way, we link marine hazards to the temperature target of the Paris Agreement and quantify avoided marine change per unit of avoided global warming. Specifically, we investigate the relationship of SAT with changes in the marine $O_2$ inventory, ocean temperature, and the metabolic index of *Deutsch et al.* (2015), the volume occupied by hypoxic water and in low latitude export production (30°S - 30°N) across the range of warming scenarios in our ensemble (Fig. 5). Distinct and often close to linear relationships emerge. Near-linearity allows us to characterize the benefits of avoided warming by single sensitivities, corresponding to the slopes of the relationships displayed in Fig. 5.

The relationships between SAT and marine hazard metrics critically depend on the time horizon considered (Fig. 5). We compare and contrast changes at the end of the 21st century, the typical assessment timescale of climate change, to peak and equilibrium changes at the millennial timescale. Peak and equilibrium changes are related to the corresponding equilibrium temperature response, while changes at the end of the 21st century are related to the transiently realized warming at the end of the 21st century. Larger magnitudes are simulated on millennial timescales compared to the near-term end of the 21st century. Assessment of ocean deoxygenation by the end of the 21st century, therefore, underestimates the full amplitude of change.

Transient (end of 21st century), peak (AD ∼3000) and equilibrium (AD ∼8000) oxygen changes exhibit distinct relationships to their corresponding warming (Fig. 5a). At the end of the 21st century, simulated oxygen decreases by 0.68 % °C$^{-1}$ of realized warming (median values). At peak oxygen decline, this sensitivity increases and oxygen decreases by 4.4 % °C$^{-1}$ of equilibrium temperature response. In other words, an avoided warming of 1°C, avoids a peak decline in marine $O_2$ inventory of 4.4%. The linear relationship breaks down for the equilibrium response. While 1.5 to 3.3 °C warming targets lead to similar and higher oxygen levels, the 9.2 °C warming target results in lower oxygen levels compared to preindustrial as discussed in the previous section. The relationships generally hold across the sampled parameter space.

The volume of low oxygen waters is particularly sensitive to warming and parameter uncertainty (Fig. 5b). We illustrate the sensitivities at the example of the volume of waters with $O_2 < 50$ mmol m$^{-3}$. At the end of the 21st century, there is a 1.7 % increase in this volume per °C of realized warming. Peak increases scale with 63 % °C$^{-1}$ of equilibrium temperature response. Uncertainties in remineralization cause a spread in this response ranging from 36-76 % °C$^{-1}$ of equilibrium temperature response (90 % confidence range): The longer the remineralization length scale, the higher this sensitivity. Pre-existing low $O_2$ waters expand and new low $O_2$ waters may develop in near bottom environments for higher forcing levels. While the lower temperature targets yield lower volumes of low oxygen waters, the 9.2 °C target yields higher low $O_2$ volumes under new steady state conditions. In brief, hypoxic waters expand over the next millennium across the scenario range and recovery

towards modern conditions is slow and in the case of high forcing incomplete. Acknowledging millennial timescales, the hazard of expanding low $O_2$ waters is much larger than when assessed on the near-term timescale.

The sensitivity of changes in low latitude export production (30°S - 30°N) is similar at the end of the 21st century and at time of peak decline. Changes scale with 2.2 % °$C^{-1}$. Export production in the low latitudes recovers for lower forcing, but remains reduced in the high forcing case.

Decreases in the metabolic index of the upper ocean scale linearly with forcing: 5.1 % °$C^{-1}$ of realized warming at the end of the 21st century and 6.2 % °$C^{-1}$ of equilibrium temperature response. Likewise, global mean oceanic temperatures increase by 0.099 °C °$C^{-1}$ of realized warming at the end of the 21st century and by 0.56 °C °$C^{-1}$ at equilibrium. In conclusion, the compound hazards related to deoxygenation and warming, as indicated by the metabolic stress index, evolve over millennia and increase with increasing anthropogenic forcing and with time.

Not only the magnitude or intensity, but also the duration of oxygen related, transient hazards, and thus the severity of the hazards increases with increasing temperature targets. The severity combines magnitude and duration of a hazard in one measure. It may be defined as the time integral of a hazard. The severity of the hazard of expanding hypoxic waters, for example, corresponds to the area under the scenario curve shown in Fig. 2h (the area enclosed by the null line and the modeled evolution, here until the end of the simulation). Fig. 2a,h, and g illustrate that the severity of the three hazards decreasing mean oxygen concentration, expanding hypoxic waters, and reduced export of particulate organic matter providing food for deep sea organisms, increases strongly from low to high temperature targets.

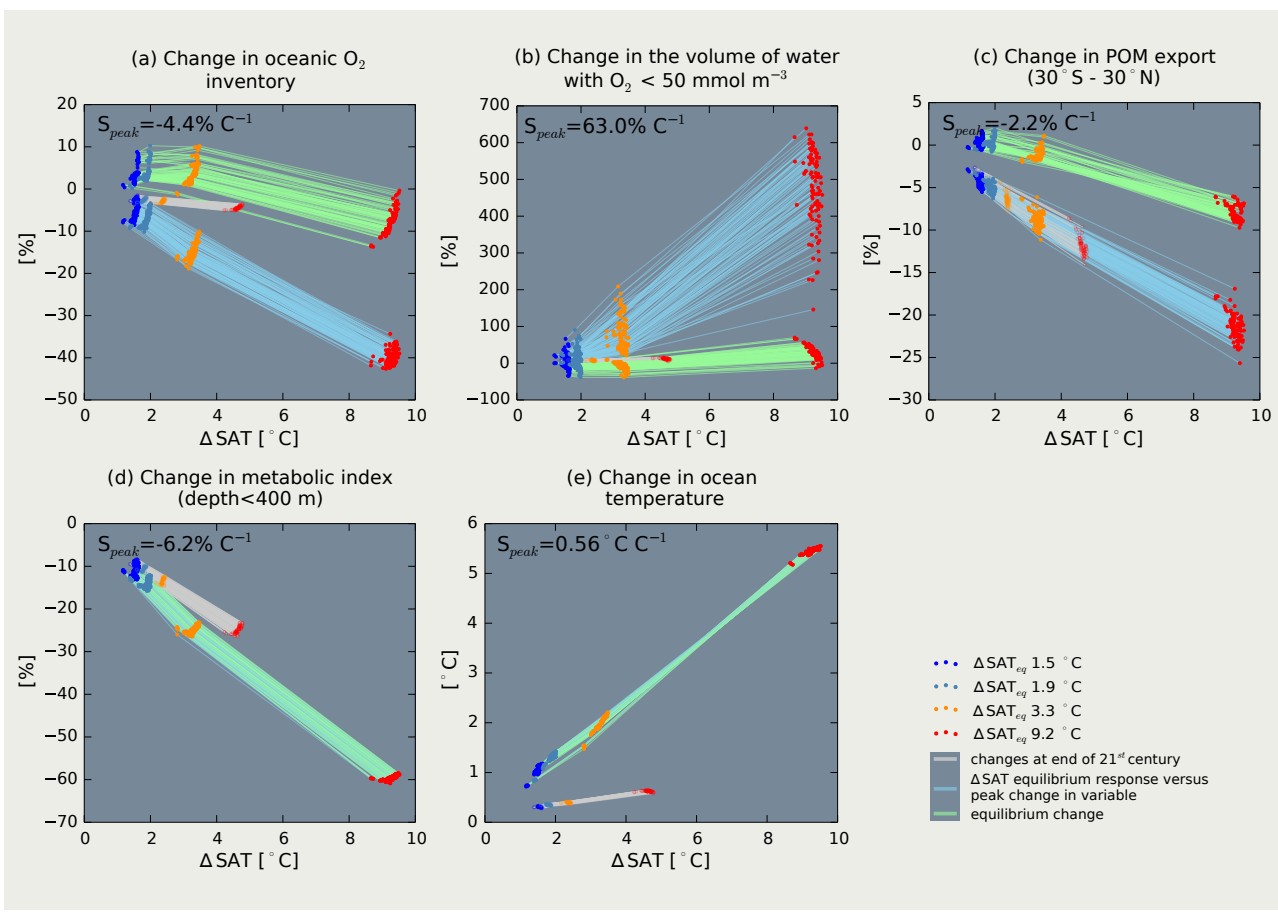

**Figure 5.** Changes in marine ecosystem stressors versus changes in global mean surface air temperature at three distinct points in time relative to 1870-1899. The colored dots indicate results of the four 100-member ensembles targeted to reach 1.5 (blue dots), 1.9 (light blue dots), 3.3 (orange dots) and 9.2 °C (red dots) warming targets. The lines connect results of individual ensemble members at 2100 AD (gray), at time of peak decline of each variable (light blue) and by the end of the simulation (light green) when a new equilibrium state has been reached. Peak and equilibrium changes in variables of interest are related to the corresponding equilibrium temperature response, while changes at the end of the 21st century are related to the transiently realized warming. Peak and equilibrium responses are indistinguishable in the figure for the metabolic index (d) and the ocean temperature change (e). $S_{peak}$ is the peak sensitivity of each variable per °C equilibrium warming.

# 5 Uncertainties in $O_2$ projections

The pattern and magnitude of simulated global $O_2$ changes are determined by the response of the overturning circulation. $O_2$ loss due to less $O_2$ solubility at higher temperatures gradually decreases oceanic $O_2$, in addition. Only few multi-millennial simulations with GCMs currently exist. The response of the overturning circulation on long timescales differs among available model simulations (including EMICs and GCMs). Uncertainties in the equilibrium climate sensitivity additionally impact projections of $O_2$ loss due to solubility. These uncertainties directly impact projections of oceanic oxygen.

Similar circulation dynamics as simluated here (Fig. 1e,f) were found by *Rugenstein et al.* (2016) based on EMIC simulations over 10,000 years with ECBILT-CLIO, which features a dynamic, quasi-geostrophic atmosphere. *Schmittner et al.* (2008), too, found similar AMOC and Indo-Pacific MOC characteristics for their EMIC UVic 2.7, which includes an atmospheric energy balance model with fixed wind fields similar to the Bern3D model, over a 2000 year simulation. *Yamamoto et al.* (2015), on the other hand, found different overturning characteristics in a simulation with a state-of-the art Earth System Model (MIROC 3.2 for a 4x$CO_2$) over 2000 years. There AMOC slowed down with no recovery, while AABW decreased only slightly and gradually increased thereafter. Predictions of AMOC have received more attention so far, and AMOC slowdown and partial or full recovery emerges in other multi-millennial simulations (*Zickfeld et al.*, 2013; *Li et al.*, 2013; *Weaver et al.*, 2012). AMOC and Southern Ocean overturning in CMIP5 Earth System Models was analyzed by *Heuzé et al.* (2015). They found AMOC and Southern Ocean overturning is positively correlated in most CMIP5 models by the end of the 21st century. Generally, preindustrial circulation states, magnitudes and timing of changes are highly model and scenario dependent such that the long-term evolution of meridional overturning is uncertain. As oxygen changes are dominated by circulation changes, this makes the oxygen prediction highly model and scenario dependent, as well. The simulated timing and strength of peak $O_2$ decrease in Bern3D is similar to what *Schmittner et al.* (2008, AD 3000, 30 % for SRES A2 high emission scenario/SAT~10 °C in Uvic 2.7) found. Other comparable simulations show earlier peaks and smaller magnitudes (*Mathesius et al.* (2015, AD 2600, 16 % decrease for RCP8.5/$\Delta$SAT~7 °C in CLIMBER-3$\alpha$), *Yamamoto et al.* (2015, after 800 model years, 10 % for 4x$CO_2$/$\Delta$SAT~8.5 °C in MIROC 3.2).

Major physical limitations of our simulations concern prescribed winds and ice-sheets. Future model studies may include sensitivity simulations with prescribed changes in the wind stress over the ocean (e.g. *Tschumi et al.*, 2008) and prescribed meltwater fluxes or apply earth system models with interactive atmospheric dynamics and ice sheets. Our study, as is the case for most climate change simulations, do not include melting of continental ice sheets, which would tend to further (transiently) reduce circulation (*Bakker et al.*, 2016) and increase the equilibrium climate sensitivity.

Current generation GCMs, such as is the case for Bern3D, have difficulty simulating the current distribution of OMZs due to missing physical processes operating at small spatial scales, such as eddies and zonal jets (*Cocco et al.*, 2013; *Bopp et al.*, 2013) or missing biogeochemical characteristics. Large model-data and model-model discrepancies exist (*Bopp et al.*, 2013).

*Laufkötter et al.* (2017) recently achieved improved representation of OMZs introducing temperature and oxygen dependence of the remineralization profile within a GCM (GFDL ESM2M). In our ensemble, the magnitude of peak increases in low $O_2$ waters depend strongly on the rate of organic matter remineralization. Temperature dependent feedback mechanisms, neglected here, may be addressed in future studies. Both particulate sinking speed and local remineralization rates, which control the remineralization profile, have been shown to be sensitive to temperature. While higher temperatures increase bacterial activity and therefore remineralization (*Bendtsen et al.*, 2014) they decrease viscosity and therefore increase sinking speed (*Taucher et al.*, 2014). The net effect on the remineralization profile is correspondingly uncertain. In addition, ecosystem structure influences the size and density of organic particles available for export (*Armstrong et al.*, 2001, 2009). Given these existing uncertainties and the coarse resolution physical models, the projections of OMZs has to be viewed with caution. Despite simulated lower background concentrations of $O_2$ in the subsurface ocean, the volumes of low $O_2$ waters decrease for steady state conditions in the model.

We neglect a number of biogeochemical feedback mechanisms that could alter biological productivity in the surface ocean and by that change remineralization fluxes in the water column. Any mechanisms that would increase remineralization would tend to decrease the oceanic oxygen, and mechanisms that decrease remineralization would increase the oceanic oxygen content. Future studies may address feedbacks from sediment interactions and imbalances from riverine input and burial (such as *Roth et al.*, 2014; *Niemeyer et al.*, 2017), temperature dependent remineralization, and variable stoichiometry. Further investigations may also address nitrogen cycle dynamics and assess the interplay of denitrification and N-fixation and of external atmospheric and terrestrial nitrogen sources.

## 6  Implications and Conclusion

In Bern3D, strong deoxygenation in all basins is projected to peak long after the end of the 21st century, and new steady state conditions establish after AD 8000 in scenarios where radiative forcing is stabilized in the next century. The equilibration timescale of oceanic oxygen is therefore longer than the thermal equilibration timescale of both the atmosphere (∼1000 years) and the ocean (∼4000 years). Based on CMIP5 models, *Sweetman et al.* (2017) discuss the deep-sea ecosystem implications of climate change by 2100. Deep sea ecosystems provide a range of services from habitat provision, nursery grounds, trophic support, refugia to biodiversity (reviewed in *Sweetman et al.*, 2017). Biogeochemical changes such as deoxygenation, warming, acidification and less food availability will likely be accompanied by exploitation of mineral resources, over fishing and dumping of pollutants and microplastics. We project largest biogeochemical changes beyond 2100 and to aggravate over millennia. How these changes will affect deep-sea ecosystems is poorly understood. The adaptation to stress may be limited by slow growth rates and long generation times of deep sea ecosystems (*Sweetman et al.*, 2017).

Figure 6a contrasts near-term (A.D. 2100) and peak changes (relative to 1870-1899) in measures of metabolically viable habitats in the upper ocean, hypoxia, and food availability as projected by Bern3D for a 1.5°C warmer world. Export in low latitudes (30°S - 30°N) as an indicator of food availability is reduced by maximally 4% over the course of the simulation in this scenario. Median decreases in the metabolic index, representing viable habitat reductions of the upper ocean, amount to 11 % for a 1.5° warmer world. The volume of low oxygen waters is particularly sensitive to anthropogenic warming and peak changes occur after the end of the 21st century. The volume of water with $O_2 < 50$ mmol m$^{-3}$ changes by 6.6 % by the end of the century and by 14 % at its peak. Meeting the 1.5°C climate target of the Paris Agreement requires very fast and very stringent emission reductions (*Steinacher et al.*, 2013; *Sanderson et al.*, 2016; *Millar et al.*, 2017). Estimates by *Steinacher et al.* (2013) for a range of scenarios show that post-2017 allowable carbon emissions from fossil fuel need to be lower than 320 GtC to meet the 1.5°C target with 66% probability (the 320 GtC are derived by adjusting the emission limit as displayed in Fig. 4 of *Steinacher et al.* (2013) for year 2000 using fossil emissions published by the Global Carbon Project). This corresponds in the most optimistic scenario to only slightly more than three decades of current fossil fuel use. The Nationally Determined Contribution, outlining emission mitigation actions by the Parties of the Paris Agreement, need to be strengthened in ambition and scope to meet the 1.5°C or the 2°C target (*Joeri et al.*, 2016). Such efforts would lead not only to lower warming compared to the current emission trajectory, but also have the benefit of reduced marine hazards as investigated here (Fig. 6b).

Higher temperature targets increase the hazard of ecosystem impacts as expressed in the investigated variables. In particular, measures of peak hypoxia exhibit a strong sensitivity to additional warming (Fig. 6b). Measures of deoxygenation, marine food scarcity, and marine aerobic habitat reduction are aggravated for the 2°C compared to the 1.5°C temperature target and investigated hazards are strongly amplified in a world where surface air temperature is stabilized at 3.3°C (Fig. 6b). Unbounded use of carbon emissions from existing fossil resources is projected not only to lead to a global warming of order 10°C (Fig. 1a,

Fig. 5; *Randerson et al.*, 2015; *Zickfeld et al.*, 2013), but also to a peak reduction in global mean oxygen inventory by almost a factor of two (Fig. 2a, Fig. 5a).

We find close to linear relationship between impact-relevant marine hazards and global mean surface air temperature. This allows us to quantify avoided hazards per unit of avoided global warming. For example, emission mitigation measures would help to reduce peak $O_2$ loss by 4.4 % $°C^{-1}$ of avoided equilibrium warming.

The Earth system response timescale to climate change spans several millennia such that anthropogenic perturbations to greenhouse-gas concentrations commit the Earth system to long-term, irreversible climate change (*Clark et al.*, 2016). Our simulations show that the long-term fate of oceanic oxygen is characterized by an initial decline followed by a recovery phase. Peak decline and associated potential adverse ecosystem impacts are projected long after stabilization of radiative forcing in the atmosphere. This adds to the list of long-term Earth System commitments including warming, acidification and sea-level rise assessed elsewhere (*Eby et al.*, 2009; *Ridgwell and Schmidt*, 2010; *Lord et al.*, 2016; *Pfister and Stocker*, 2016; *Steinacher and Joos*, 2016; *Clark et al.*, 2016). Long-term, multi-millennial perspectives are thus required for a full account of climate-related ocean risks.

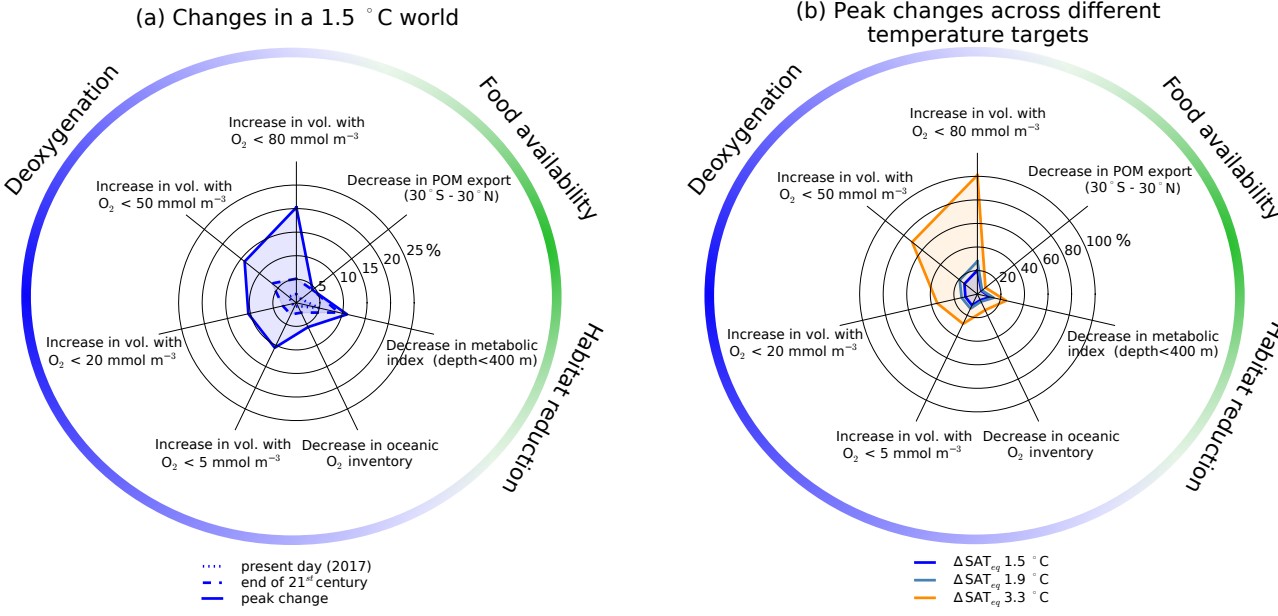

**Figure 6.** Contrasting hazards of ecosystem impacts expressed in measures of hypoxia, metabolic viability of the upper ocean, and food availability. a) Changes for a 1.5 °C warmer world at present, at the end of the century and compared to peak changes. b) Peak changes across 1.5, 1.9 and 3.3 °C temperature targets. Lines correspond to the median response across each ensemble relative to 1870-1899.

*Data availability.* Model output is available upon request to the corresponding author (battaglia@climate.unibe.ch).

**Appendix A:  Spatial properties of a representative ensemble member reaching a 1.5°C warming target**

In this appendix we document additional spatial properties of the representative ensemble member reaching a 1.5°C warming target. Figure A1 illustrates the meridional overturning streamfunction for PI, year A.D. 2050 where the AMOC is at its lowest value, and for new steady state conditions. Figure A2 illustrates the evolution of temperature, salinity, and density anomalies across a transect from the Atlantic Ocean, through the Southern Ocean and into the Pacific at A.D. 2100, A.D. 3150 when the $O_2$ inventory is at its lowest values, and for new steady state conditions. Figure A3 shows the anomalies in total $O_2$ and the contributions from biology and solubility components for new steady conditions relative to PI. In addition, anomalies in ideal age are shown.

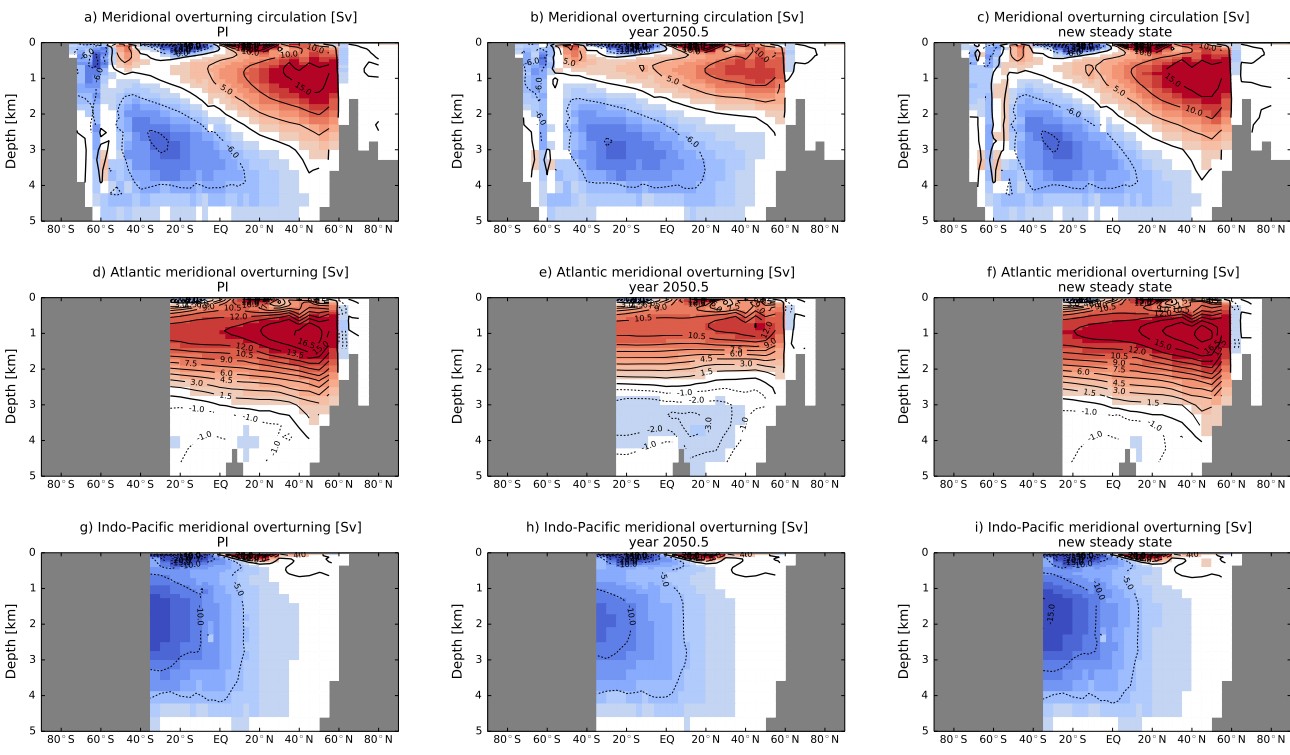

**Figure A1.** Meridional overturning streamfunction in (a-c) the world ocean, (d-f) Atlantic ocean, and (g-i) Indo-Pacific for PI, year 2050, and for new steady state conditions for a representative ensemble member reaching a 1.5°C warming target (columns). Circulation is clockwise along positive (red) contours and anticlockwise along the negative (blue) contours.

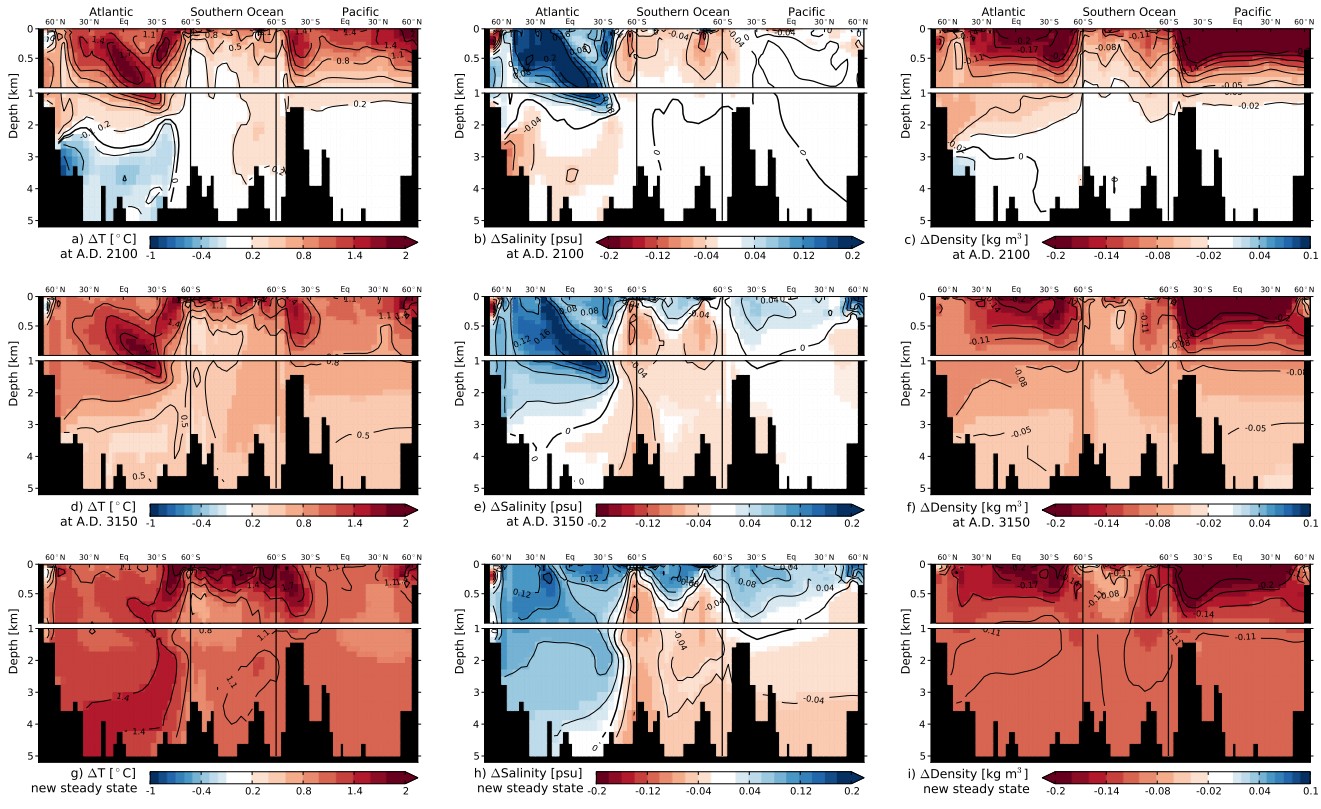

**Figure A2.** Changes in temperature, salinity and density at A.D. 2100 (a,b,c), at A.D. 3150 (d,e,f), and for new steady state conditions (g,h,i) compared to pre-industrial conditions.

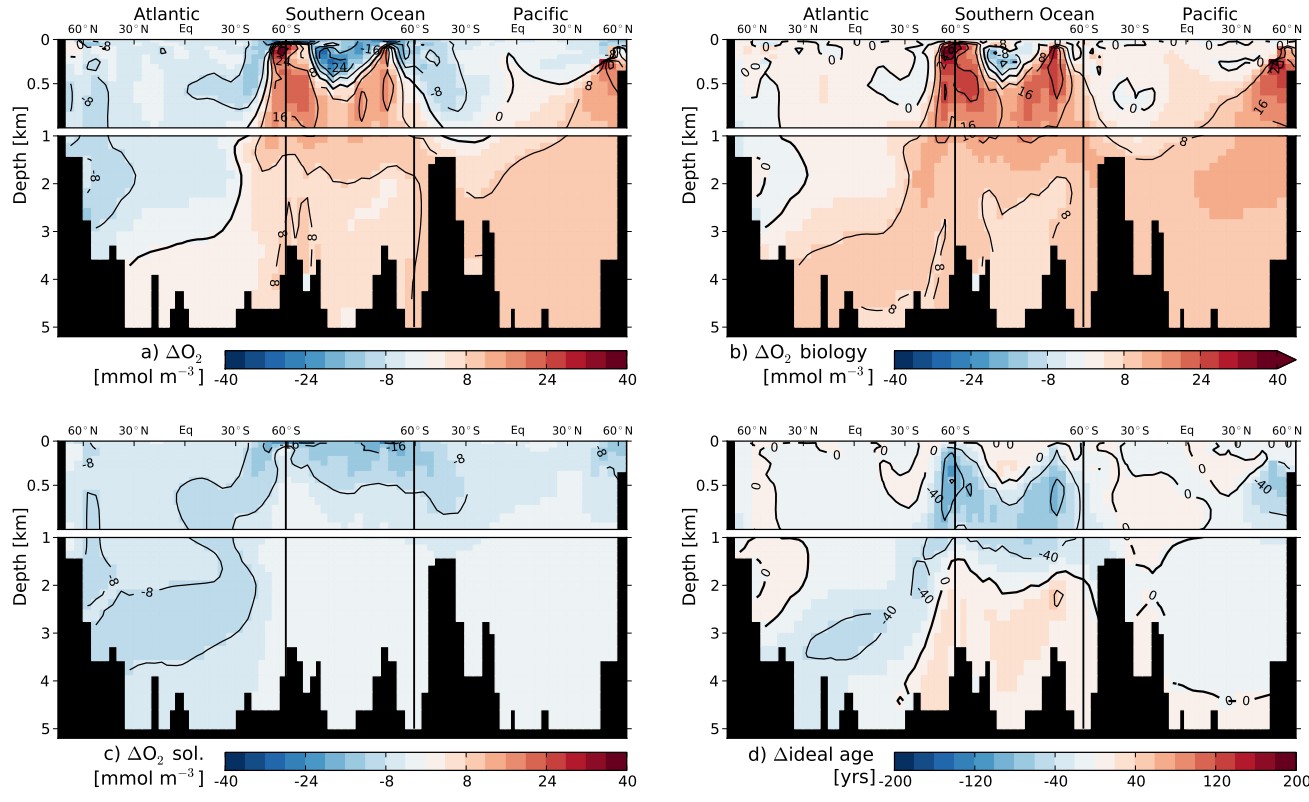

**Figure A3.** Changes in O₂ and its components for new steady state conditions relative to preindustrial steady state for a single, representative ensemble member reaching a 1.5 °C warming target. a) Change in total O₂, b) change in O₂ due to biology, c) change in O₂ due to solubility, d) changes in ideal age.

*Competing interests.* The authors declare that they have no conflict of interest.

*Acknowledgements.* We thank Andreas Oschlies and Andreas Schmittner for their thorough reviews of our manuscript, which helped imporve the communication of our results. We also thank Patrik Pfister, and Thomas Frölicher for fruitful discussions. This work was supported by the Swiss National Science Foundation (200020‗172476).

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
