# Peer review of "Hazards of decreasing marine oxygen: the near-term and millennial-scale benefits of meeting the Paris climate targets"

_Earth System Dynamics, 2017_

## Referee Comment (RC1) · Anonymous Referee #1 · 19 Nov 2017

The manuscript describes results of a number of simulations with an Earth system model of intermediate complexity with respect to changes in oceanic oxygen content and specific ecosystem stressors, such as the volume of low-oxygen waters and the value of a metabolic index. The authors present a number of interesting findings, for example that deoxygenation peaks about a thousand years after stabilization of radiative forcing and oxygen recovers thereafter. It is, however, difficult to identify a main message. The benefits of meeting the Paris targets is mentioned in the title, but the manuscript quickly leaves this storyline, with no mentioning of the Paris climate goals after the introduction. Also, there is little information provided on shorter than millennial timescales - i.e. the near-term goal mentioned in the title is not discussed in the

manuscript.

The discussion of uncertainties is limited to parameter uncertainties. However, systematic shortcomings of the intermediate complexity model, such as fixed winds and ice sheets or the neglect of sediments and nitrogen cycle feedbacks. These shortcoming may be much larger than those discussed in the manuscript. This needs to be discussed.

Overall, there is substantial new and interesting science in the work presented, but in addition to the absence of a clear storyline, the presentation is very descriptive and does not go into sufficient depth to really explain the interesting findings. I don't think the manuscript is ready for publication in its present form. Instead, the manuscript requires a major reorganization, possibly a new title and clearly a well-defined storyline.

Individual comments:

p.2, l.5 what is the justification for calling this 'is now a key scientific task'? For what? Why should people be interested on timescales of several millennia?

p.2, l.11 & 17. Hypoxia is defined, then suboxia is used. What is the difference (if any)? Why are different terms used?

p.3 l.28 Does this mean that winds are unchanged during the 8000yr global warming simulations? What are the implications of this? Could this explain the systematic differences with respect to paleo inferences about oxygen changes under global warming? I think this requires a detailed discussion.

section 2.3 The model evaluation is presented in a manuscript under review and not available to the reviewer/reader right now. Impossible to judge. I suggest to include maps and profiles of oxygen distributions in this manuscript.

p.6, l.15. 'deeper' —> longer? This suggests that low-oxygen waters are simulated mostly in the deep ocean, whereas in reality they are located at a depth of a few hundred meters, so that the real-ocean low oxygen volumes should be more sensitive

for shorter remineralization length scales. This requires some explanation.

p.6, l.20ff Why is the recovery level for export so similar, and that for oxygen so different among the models?

p.6, l. 24 & 26 Why does the metabolic index scale linearly with forcing (i.e. equilibrium temperature) when it changes non-linearly with temperature?

p.9 l.2 Why is this representative?

section 5. It would be good to learn more about the critical factors that determine model-model differences in simulated changes and recovery of circulation and oxygen. e.g. model resolution? treatment of wind forcing? different biogeochemical assumptions? temperature effects on remineralization?

p.14, l.5ff For which year are the changes given?

---

## Referee Comment (RC2) · Anonymous Referee #2 · 2 Dec 2017

The authors describe results from a modeling study projecting long-term future ocean oxygen evolution for different carbon emission scenarios. As such it is within the scope of ESD. It is one of a relatively few studies that go beyond the centennial time scale and that consider millennial and multi-millennial timescales. I'm not sure which scientific question(s) the paper addresses. If there is one, or several, it may be useful to make this clearer in the introduction. Its title indicates that investigations are centered around assessment of benefits from the Paris agreement.

I have mixed feelings about the manuscript. There are certainly novel aspects. For example, calculations of a metabolic index or diagnostics of relationships between oxy-

gen related changes and global mean equilibrium temperature. These may be useful for other scientists or policy makers.

On the other hand, there are statements (in the abstract, introduction, and conclusions) that sound like novel achievements but that in fact are not new and have been documented before (e.g. the long timescale for deep ocean oxygen changes). Another irritation to me were the short discussions of paleo oxygen changes in relation to the future projections presented. The paleo oxygen changes are a complex issue by themselves and I did not find the cursory discussion provided here helpful. There is substantial evidence that the glacial-interglacial changes were influenced by iron fertilization or some other biogeochemical process that increased macro-nutrient utilization during glacial periods (e.g. Schmittner and Somes, 2016, Paleoceanogr., doi: 10.1002/2015PA002905), something that is not considered in the future projection simulations discussed in this paper. This makes even a qualitative comparison difficult if not impossible. Moreover, large changes in ice sheets and sea level occurred during glacial-interglacial changes, which are not considered here either.

The paper is sparingly illustrated and includes many statements that are not supported by evidence or figures. E.g. the authors claim they have separated different contributions to the oxygen changes (production, consumption, solubility), but not a single figure is shown illustrating those.

Even though the authors acknowledge that many processes are not considered in their projections (page 4, lines 26-28) they do not discuss the possible impacts of those omissions on their results. E.g. the large increases in suboxic zones projected for high emission scenarios will lead to increased denitrification and a reduced fixed nitrogen inventory, which will affect productivity on long timescales (e.g. Schmittner et al., 2008). On long timescales, we would expect ice sheets to change considerably (at least for the high emission scenarios).

Another weakness of the manuscript is that in many instances model responses are

simply described but not explained or understood. My notes include lots of "why?" annotations as listed below.

Specific comments:

Title: The "near-term" does not seem to be a focus of the manuscript. The term is not mentioned anywhere else in the text.

Abstract lines 6-7: "Deoxygenation ... forcing." This is not a new finding and has been shown before, e.g. in Schmittner et al. (2008). Page 4 line 9: "production, consumption, solubility" results of this decomposition are not shown in the remainder of the manuscript

Page 4 line 16-17: "We show that the oceanic oxygen equilibration timescale is considerably longer than its thermal equilibration timescale". The long oxygen equilibration timescale has been shown before (e.g. Schmittner et al. 2008). Perhaps more of a discussion of previous long-term studies (the ones cited in the previous sentence) would be useful to better understand what is new and what is not.

Page 5 line 12: Battaglia and Joos (2017) is not available

Page 5 lines 16-17: please define the quoted variables precisely. What precisely is the AMOC index? How was it calculated? The same for the Indo-Pacific overturning and export production.

Page 5 lines 26-29: This is not new. It has been shown before in Schmittner et al. (2008).

Page 6 line 7: Why do the lower emission scenarios lead to increased oxygen?

Page 6 lines 9-10: Why do lower mixing coefficients lead to larger decreases in oxygen?

Page 7 lines 5,6: are the production and consumption tracer results shown somewhere?

Page 7 lines 19-20: why do higher forcing levels lead to these MOC changes?

Page 9 line 4: Why are subsurface ages younger?

Page 9 line 9: increased stratification is not shown. Is it really increased at equilibrium or is this just a transient effect? If it is increased is this due to temperature or salinity? Fig. 3 indicates that at least in the Atlantic stratification is not increased due to temperature although export production there is decreased.

Page 9 line 14: Why does the temperature anomaly develop there?

Page 9 lines 18-20: Is this shown somewhere?

Page 10: Part of the figure caption is missing.

Page 12 lines 4-5: what are these numbers based on?

Page 12 line 7: "comparatively strong" compared to what?

Page 12 lines 27-29: The discussion here is too simplistic. In the paleodata the deep ocean's oxygen increased while it decreased in the thermocline. I don't see evidence provided that this is similar to the model data. It is not similar to Fig. 3a, rather the opposite, I would say. Page 12 lines 30-31: I don't agree with the statement "Proxies of past ocean oxygenation and ventilation reveal similar structural changes and mechanisms." Increased nutrient utilization e.g. from iron fertilization also most likely played a role in glacial-interglacial changes (e.g. Schmittner and Somes, 2016, Paleoceanogr., doi:10.1002/2015PA002905).

Page 12 line 31: It is not clear if the overturning increased. Changes in overturning strength remain controversial (e.g. Kurahashi-Nakamura et al., 2016, Paleoceanogr. doi:10.1002/2016PA003001).

---

## Author Comment (AC1) · 8 Feb 2018

Please find our replies to all review comments in the supplementary.

Please also note the supplement to this comment:
https://www.earth-syst-dynam-discuss.net/esd-2017-90/esd-2017-90-AC1-supplement.zip

———————————————

---

## Author Response (AR1)

Reply to reviews for the manuscript

**Hazards of decreasing marine oxygen: the near-term and millennial-scale benefits of meeting the Paris climate targets**

submitted to Earth Syst. Dynam. Discuss.
by Gianna Battaglia, Fortunat Joos

February 8, 2018

We thank the two anonymous reviewers for their reviews and constructive comments. We much appreciate the effort and time committed by the reviewers.

We appreciate their main concerns. As a result, we reorganized the presentation of our results and provide additional information and additional figures as requested. Also, we remove the discussion on paleo oxygen changes as requested. The introduction and conclusion are largely re-written to strengthen the main messages.

Please find below our response to the comments by the reviewers and suggested text additions to the manuscript. A new manuscript, and a manuscript version with track changes is attached.

**Anonymous Referee #1**

The manuscript describes results of a number of simulations with an Earth system model of intermediate complexity with respect to changes in oceanic oxygen content and specific ecosystem stressors, such as the volume of low-oxygen waters and the value of a metabolic index. The authors present a number of interesting findings, for example that deoxygenation peaks about a thousand years after stabilization of radia- tive forcing and oxygen recovers thereafter. It is, however, difficult to identify a main message. The benefits of meeting the Paris targets is mentioned in the title, but the manuscript quickly leaves this storyline, with no mentioning of the Paris climate goals after the introduction. Also, there is little informa- tion provided on shorter than millen- nial timescales - i.e. the near-term goal mentioned in the title is not discussed in the manuscript.

In this manuscript, we compare and contrast the usual assessment timescale of climate change at the end of the 21st century, here defined as near-term, to the multi-millennial equilibration timescale of ocean biogeochemistry. The original Figure 1 and the original Figure 4 contrasted these near-term changes to the peak changes as simulated by our model. Also, all timeseries plots visually highlight the near-term timescale (A.D. 2100) to the multi-millennial equilibration timescale (A.D. 10,000). We clarify this point in the introduction by adding the following paragraph. In addition, we add more explicit mentioning of the Paris Agreement on various places in the manuscript.

Given the long residence time of anthropogenic $CO_2$ in the atmosphere, and long equilibration timescales of the ocean overturning circulation, anthropogenic climate change will grow and persist beyond the end of the 21st century, the typical near-term assessment timescale of climate change (*Clark et al.*, 2016). Only few studies have assessed ocean biogeochemistry and the oceanic oxygen content beyond this near-term timescale. Available studies employ a range of physical and biogeochemical complexity levels from box models to general circulation models (GCMs). Oxygen concentrations are simulated to decline beyond the 21st century on multi-centennial timescales (*Matear and Hirst*, 2003; *Hofmann and Schellnhuber*, 2009; *Mathesius et al.*, 2015). Siumlations covering two millennia show a recovery phase thereafter (*Schmittner et al.*, 2008; *Yamamoto et al.*, 2015). In most studies, simulated oxygen concentrations have not reached new steady state

conditions at the end of the simulation. Low order Earth system models and Earth System Models of Intermediate complexity integrated by up to 100,000 years have demonstrated the potential for long-term ocean oxygen depletion in response to carbon dioxide emissions and the long equilibration time scales of ocean biogeochemical variables in response to carbon emissions (*Shaffer et al.*, 2009; *Ridgwell and Schmidt*, 2010). Multi-millennial simulations are therefore required to assess the full amplitude of ocean biogeochemical changes and new steady state conditions due to anthropogenic climate change.

The discussion of uncertainties is limited to parameter uncertainties. However, systematic shortcomings of the intermediate complexity model, such as fixed winds and ice sheets or the neglect of sediments and nitrogen cycle feedbacks. These shortcom- ing may be much larger than those discussed in the manuscript. This needs to be discussed.

We add this discussion to section 5 Uncertainties in $O_2$ projections:

Major physical limitations of our simulations concern prescribed winds and ice-sheets. Future model studies may include sensitivity simulations with prescribed changes in the wind stress over the ocean (e.g. *Tschumi et al.*, 2008) and prescribed meltwater fluxes or apply earth system models with interactive atmospheric dynamics and ice sheets. Our study, as is the case for most climate change simulations, do not include melting of continental ice sheets, which would tend to further (transiently) reduce circulation (*Bakker et al.*, 2016) and increase the equilibrium climate sensitivity.

We neglect a number of biogeochemical feedback mechanisms that could alter biological productivity in the surface ocean and by that change remineralization fluxes in the water column. Any mechanisms that would increase remineralization would tend to decrease the oceanic oxygen, and mechanisms that decrease remineralization would increase the oceanic oxygen content. Future studies may address feedbacks from sediment interactions and imbalances from riverine input and burial (such as *Roth et al.*, 2014; *Niemeyer et al.*, 2017), temperature dependent remineralization, and variable stoichiometry. Further investigations may also address nitrogen cycle dynamics and assess the interplay of denitrification and N-fixation and of external atmospheric and terrestrial nitrogen sources. The resulting impact on the fixed nitrogen inventory in the ocean are currently unclear.

Overall, there is substantial new and interesting science in the work presented, but in addition to the absence of a clear storyline, the presentation is very descriptive and does not go into sufficient depth to really explain the interesting findings. I dont think the manuscript is ready for publication in its present form. Instead, the manuscript requires a major reorganization, possibly a new title and clearly a well-defined storyline.

> We follow the suggestions of the reviewers and change the presentation of the results. We now show and describe first Figure 2, the temporal evolution of critical variables, and Figure 3, spatial changes in ecosystem stressors at peak decline. These are followed by Figures 1 and 4. In addition, we provide additional details on mechanisms following reviewer #2. We add additional variables as timeseries plots and include further variables as section plots for process attribution and discuss now mechanisms of change in greater detail.
>
> The new organization, and more explicit mentioning of the Paris climate goals should justify the choice of our title.

Individual comments:

p.2, l.5 what is the justification for calling this is now a key scientific task? For what? Why should people be interested on timescales of several millennia?

> The Parties to the UNFCCC are '*determined to protect the climate system for present and future generations*' and the UNFCCC mentions '*the threats of irreversible damage*' in its Article 3. The parties of the follow-up Paris Agreement recognize '*the need for an effective and progressive response to the urgent threat of climate change on the basis of the best available scientific knowledge*', and the Agreement notes '*the importance of ensuring the integrity of all ecosystems, including Oceans*'. It remains thus an important task to further the scientific knowledge on the impacts of global warming on the ocean, considering potentially irreversible and long-term changes that may harm ocean ecosystems and threaten the well- being of future generations. To this end, we project the response in oceanic oxygen content and a number of additional ecosystem stressors including warming, export production and a metabolic index for a range of warming targets, including the 1.5 and 2 °C targets mentioned in the Paris Agreement. We consider within the Bern3D Earth System Model of Intermediate Complexity (EMIC) changes

over this century as well as long-term changes over the next 8000, recognizing the multi-millennial equilibration time scales of marine biogeochemical cycles.

The specific wording has been removed from the introduction in the course of the largely re-written introduction. The underlying reasoning is still conveyed.

p.2, l.11 & 17. Hypoxia is defined, then suboxia is used. What is the difference (if any)? Why are different terms used?

Hypoxia and suboxia refer to different $O_2$ concentration ranges. Hypoxia, defined as $O_2 < 50$ mmol m$^{-3}$, refers to conditions leading to $O_2$-stress for many macroorganisms. Suboxia refers to lower $O_2$ concentrations, here defined as $< 5$ mmol $O_2$ m$^{-3}$, leading to anaerobic metabolism.

Text clarified as:

> $O_2$ is vital for aerobic organisms in the ocean and typical thresholds leading to $O_2$-stress for many macroorganisms (hypoxia) are around 50 mmol $O_2$ m$^{-3}$.
>
> Suboxic ($<5$ mmol $O_2$ m$^{-3}$) or anaerobic conditions can also lead to production of poisonous $H_2S$ within sediments

p.3 l.28 Does this mean that winds are unchanged during the 8000yr global warming simulations? What are the implications of this? Could this explain the systematic differ- ences with respect to paleo inferences about oxygen changes under global warming? I think this requires a detailed discussion.

We removed the paleo discussion and added text on potential implications of wind changes in section 5 as detailed further above.

section 2.3 The model evaluation is presented in a manuscript under review and not available to the reviewer/reader right now. Impossible to judge. I suggest to include maps and profiles of oxygen distributions in this manuscript.

Apologies. The manuscript is now available from GBC:

http://dx.doi.org/10.1002/2017GB005671, DOI: 10.1002/2017GB005671

p.6, l.15. deeper  T¿ longer? This suggests that low-oxygen waters are simulated mostly in the deep ocean, whereas in reality they are located at a depth of a few hundred meters, so that the real-ocean low oxygen volumes should be more sensitive for shorter remineralization length scales. This requires some explanation.

We change 'deeper' to 'longer'.

We do not suggest that low oxygen waters are simulated in the deep ocean under modern conditions. The paragraph concerns the global warming experiments. The Bern3D simulates low $O_2$ waters in the thermocline of the modern ocean as observed (see Figure 3, Figure 7a, Table D1 of *Battaglia and Joos*, 2018). This is mentioned in section '2.3 Pre-Industrial characteristics'.

We expand the text on the volume of low $O_2$ waters in section 3:

Oxygen-poor waters ($O_2 < 50$ mmol m$^{-3}$, Fig. 2h) are simulated to transiently increase across all scenarios. The response is characterized by high uncertainty as introduced by the sampled parameters. Under new equilibrium conditions, the volume of low $O_2$ waters is reduced for low and intermediate forcing and remains higher than pre-industrial in the high forcing case.

Turning to uncertainties in our perturbed parameter ensemble, we find that variations in the vertical diffusion parameter ($k_{diff-dia}$) dominate the uncertainty in the globally-averaged evolution of ideal age, sea ice cover, temperature and $O_2$. The modeled uncertainty in the volume of low $O_2$ waters is dominated by different values of the $\alpha_{\mathrm{aerob}}$ parameter. Whether a threshold in $O_2$ concentration is met depends on the pre-industrial tracer distribution. Longer remineralization length scales bring more remineralization to depth, leading to higher $O_2$ consumption.

p.6, l.20ff Why is the recovery level for export so similar, and that for oxygen so different among the models?

We amend the paragraph on export changes with the following explanation in the manuscript. Please also refer to Figure 1 here, illustrating the export anomalies across three scenarios for new steady state conditions relative to preindustrial.

Global export production is simulated to decline over the first few centuries, and reach higher values under new steady state conditions (Fig. 2g). The decline is stronger for higher forcing, while the recovery level of global export production is similar across the scenarios. Bern3D transiently simulates decreased export in the mid- and low latitudes (Fig. 4c, see also *Steinacher et al.* (2009); *Battaglia and Joos* (2018)) as a result of increased stratification (Fig. A2c,f,i) and reduced nutrient concentrations in the surface ocean (Fig. 4b). In the high latitudes, the model simulates increased export production, as a result of less temperature and light limitation as surface waters warm and sea ice retreats. This pattern of decreased export in mid- and low latitudes and increased export in high latitudes is similar across the scenarios. Export production in the low latitudes fully recovers for lower forcing and partially recovers for higher forcing. The lower recovery level in the low latitudes is compensated by higher increases in the high latitudes for high forcing. The magnitude of positive and negative changes increases with forcing, but the global anomalies remain comparable at the end of the simulation.

[Figure]

Figure 1: Export anomalies for new steady state conditions relative to preindustrial for the representative ensemble member and three different scenarios reaching 1.5, 3.3 and 9.2°C warming targets.

p.6, l. 24 & 26 Why does the metabolic index scale linearly with forcing (i.e. equilibrium temperature) when it changes non-linearly with temperature?

The metabolic index changes as a result of changes in T and $O_2$ (see Figure 4 of the new manuscript). The metabolic index as a function of temperature can be approximated linearly from 0-15°C, and by another linear function in the temperature range from 15-35°C.

We add a sentence to section 2.1:

One may note that the exponential curve varies approximately linearly for typical global warming associated temperature changes as $E_0/k_b(\approx 10{,}000 \text{ K})$ is large.

p.9 l.2 Why is this representative?

It is a member with parameter values close to the standard/median parameter values. We add the following text to section 2.2 Ensemble and Scenarios:

A normal distribution is used to sample $\alpha_{\text{aerob}}$ with a standard value of -0.83 and a standard deviation of -0.0625. $\alpha_{\text{denit}}$ is sampled uniformly between -0.1 and -0.01. And a lognormal distribution is used to sample $k_{diff-dia}$ (standard value=2.25E-5 $\text{m}^2 \text{ s}^{-1}$, shape paramter=0.2, location parameter=0). We choose a single ensemble member with parameter values close to the standard values as representative ensemble member to illustrate spatial anomalies ($\alpha_{\text{aerob}}$=-0.85, $\alpha_{\text{denit}}$=-0.037, $k_{diff-dia}$=2.05E-05 $\text{m}^2 \text{ s}^{-1}$).

section 5. It would be good to learn more about the critical factors that determine model-model differences in simulated changes and recovery of circulation and oxygen. e.g. model resolution? treatment of wind forcing? different biogeochemical assump- tions? temperature effects on remineralization?

We now provide additional information characterizing the models and better distinguish EMCIS from state-of-the art GCMs. We do not have the information to reliably assess the model-model differences and their influence on the simulated changes in $O_2$ and circulation.

p.14, l.5ff For which year are the changes given?

Thanks. We clarify the text by:

Figure 6a contrasts near-term (A.D. 2100) and peak changes (relative to 1870-1899) in measures of metabolically viable habitats in the upper ocean, hypoxia, and food availability as projected by Bern3D for a 1.5° warmer world. Export in low latitudes (30°S - 30°N) as an indicator of food availability is reduced by maximally 4% over the course of the simulation in this scenario. Median decreases in the metabolic index, representing viable habitat reductions of the upper ocean, amount to 11 % for a 1.5° warmer world.

**Anonymous Referee #2**

The authors describe results from a modeling study projecting long-term future ocean oxygen evolution for different carbon emission scenarios. As such it is within the scope of ESD. It is one of a relatively few studies that go beyond the centennial time scale and that consider millennial and multi-millennial timescales. Im not sure which scientific question(s) the paper addresses. If there is one, or several, it may be useful to make this clearer in the introduction. Its title indicates that investigations are centered around assessment of benefits from the Paris agreement.

> The introduction is largely re-written. Please refer to the attached manuscript. In particular, we add:
>
> > In this study, we assess the effectiveness of the Paris climate targets in reducing hazards of decreasing oceanic oxygen, ocean warming and marine export productivity as simulated by the Bern3D Earth system model of intermediate complexity. To this end, we prescribe in the model four different scenarios where anthropogenic GHG forcing is stabilized by 2300 AD either under stringent mitigation limiting equilibrium global surface air warming to 1.5 or 2°C above preindustrial or following business-as-usual 21th century emissions. Simulations are run to year AD 10,000 by which time the ocean has reached new steady state conditions. This allows us assess reversibility and the full amplitude of changes, which are larger than the near-term changes at the end of the 21st century. We summarize the outcomes developing global metrics which quantify avoided marine hazards per avoided global warming.

I have mixed feelings about the manuscript. There are certainly novel aspects. For example, calculations of a metabolic index or diagnostics of relationships between oxygen related changes and global mean equilibrium temperature. These may be useful for other scientists or policy makers.

On the other hand, there are statements (in the abstract, introduction, and conclu- sions) that sound like novel achievements but that in fact are not new and have been documented before (e.g. the long timescale for deep ocean oxygen changes).

It is not our intention to make unjustified claims. We aimed to cite the relevant literature on long- term $O_2$ changes in the introduction and cited on p3, l.15: *Yamamoto et al.* (2015); *Matear and Hirst* (2003); *Schmittner et al.* (2008); *Mathesius et al.* (2015). We further discuss the findings of these and related studies in comparison with our results in section 5 (p. 12, line 10 to 24). Following the request of the reviewer, we provide a brief summary of the key finding of earlier long-term projections to allow the reader to better put our results in context of previous work. Please see the reply to reviewer #1 or the revised manuscript for the exact wording.

Another irritation to me were the short discussions of paleo oxygen changes in relation to the future projections presented. The paleo oxygen changes are a complex issue by themselves and I did not find the cursory discussion provided here helpful. There is substantial evidence that the glacial-interglacial changes were influenced by iron fertilization or some other bio-geochemical process that increased macro-nutrient uti- lization during glacial periods (e.g. Schmittner and Somes, 2016, Paleoceanogr., doi: 10.1002/2015PA002905), something that is not considered in the future projection sim- ulations discussed in this paper. This makes even a qualitative comparison difficult if not impossible. Moreover, large changes in ice sheets and sea level occurred during glacial-interglacial changes, which are not considered here either.

We acknowledge this point and remove the paleo discussion.

The paper is sparingly illustrated and includes many statements that are not supported by evidence or figures. E.g. the authors claim they have separated different contri- butions to the oxygen changes (production, consumption, solubility), but not a single figure is shown illustrating those.

The original Figure 2e showed the explicit $O_2$ solubility term. The biological imprint of $O_2$ changes is given implicitly by the difference of the total $O_2$ tracer (Figure 2b) and the solubility tracer (Figure 2e).

We now provide new Figures illustrating the changes arising from the four $O_2$ tracers (total, solubility, utilization, production) as timeseries and as section plots (new Figure 2a,b,c,e, new Figure 3, new Figure A3).

Even though the authors acknowledge that many processes are not considered in their projections (page 4, lines 26-28) they do not discuss the possible impacts of those omissions on their results. E.g. the large increases in suboxic zones projected for high emission scenarios will lead to increased denitrification and a reduced fixed nitrogen inventory, which will affect productivity on long timescales (e.g. Schmittner et al., 2008). On long timescales, we would expect ice sheets to change considerably (at least for the high emission scenarios).

> We now discuss these points in the discussion (5 Uncertainties in $O_2$ projections). It remains unclear whether the projected $O_2$ changes may lead to a reduced inventory of fixed nitrogen, as fixation of nitrogen may also change under global warming and elevated $CO_2$. Please see reply to reviewer #1 or the revised MS for the exact wording.

Another weakness of the manuscript is that in many instances model responses are simply described but not explained or understood. My notes include lots of why? annotations as listed below.

> The primary focus of this MS submitted to the ESDD Special Issue on 'The Earth system at a global warming of 1.5°C and 2.0°C' is to document changes in measures of marine oxygen in relation to the Paris temperature targets. We now provide additional mechanistic explanations as requested. See answers to specific points below.

Specific comments:

Title: The near-term does not seem to be a focus of the manuscript. The term is not mentioned anywhere else in the text.

> We clarify that 'near-term' is used for 21st century changes as assessed in most studies on climate change in the introduction. Please see our answer to reviewer #1 on this point for the exact wording

Abstract lines 6-7: Deoxygenation...forcing. This is not a new finding and has been shown before, e.g. in Schmittner et al. (2008).

> We are aware of the *Schmittner et al.* (2008) study; the study is cited 3 times in the submitted MS. We clarify in the revised MS that our projections cover a substantially longer time period than addressed in earlier simulations (8000 compared to 2000 years covered by *Schmittner et al.* (2008)).
>
> We amended the sentence:
>
>> Deoxygenation peaks about thousand years after stabilization of radiative forcing and new steady state conditions establish after AD 8000 in our model.
>
> We now discuss the findings of *Schmittner et al.* (2008) and other studies in the introduction (see answer above).

Page 4 line 9: production, consumption, solubility results of this decomposition are not shown in the remainder of the manuscript

> We added new figures (new Figure 2a,b,c,e, new Figure 3, new Figure A3) to show these results.

Page 4 line 16-17: We show that the oceanic oxygen equilibration timescale is consid- erably longer than its thermal equilibration timescale. The long oxygen equilibration timescale has been shown before (e.g. Schmittner et al. 2008). Perhaps more of a dis- cussion of previous long-term studies (the ones cited in the previous sentence) would be useful to better understand what is new and what is not.

> We discuss the finding of the *Schmittner et al.* (2008) and other studies in the introduction (see answers above).

Page 5 line 12: Battaglia and Joos (2017) is not available

> Apologies. The manuscript is now available from GBC:
>
> http://dx.doi.org/10.1002/2017GB005671, DOI: 10.1002/2017GB005671

Page 5 lines 16-17: please define the quoted variables precisely. What precisely is the AMOC index? How was it calculated? The same for the Indo-Pacific overturning and export production.
* * *
Added text:

> The maximum of the Atlantic meridional overturning streamfunction below 400 m depth (AMOC) ranges from 16.5 to 19.7 Sv. The minimum of the Indo-Pacific meridional overturning streamfunction below 400 m depth (Indo-Pacific MOC) ranges between -13.6 to -15.6 Sv. Export of particulate organic matter at 75 m ranges from 9.0 to 11.4 Gt C yr$^{-1}$.
* * *
Page 5 lines 26-29: This is not new. It has been shown before in Schmittner et al. (2008).
* * *
Now discussed in the introduction (see answers above)
* * *
Page 6 line 7: Why do the lower emission scenarios lead to increased oxygen?
* * *
The explanation on changes in Section 3 is improved. Please refer to the attached manuscript. In brief, a more vigorous circulation at the new compared to the PI steady state leads to an increase in ocean oxygen counteracted by a solubility/warming-driven reduction in oxygen.
* * *
Page 6 lines 9-10: Why do lower mixing coefficients lead to larger decreases in oxy- gen?
* * *
Thanks for this question. The statement was in fact wrong, and the opposite is true. A full attribution of physical changes is beyond the scope of the manuscript. The statement is removed from the manuscript and instead, we add the following to section 3 Marine changes in temperature, circulation and biogeochemistry:

> Turning to uncertainties in our perturbed parameter ensemble, we find that variations in the vertical diffusion parameter ($k_{diff-dia}$) dominate the uncertainty in the globally-averaged evolution of ideal age, sea ice cover, temperature and $O_2$. The modeled uncertainty in the volume of low $O_2$ waters is dominated by different

> values of the $\alpha_{\text{aerob}}$ parameter. Whether a threshold in $O_2$ concentration is met depends on the pre-industrial tracer distribution. Longer remineralization length scales bring more remineralization to depth, leading to higher $O_2$ consumption.

Page 7 lines 5,6: are the production and consumption tracer results shown somewhere?

> We added new figures (new Figure 2a,b,c,e, new Figure 3, new Figure A3).

Page 7 lines 19-20: why do higher forcing levels lead to these MOC changes?

> We improved section 3 and show additional physical variables in additional figures in the main text and the appendix (sea-ice, streamfunction, temperature, salinity, density) for transparency and discuss these physical changes and their relationship. Please see section 3 in the revised manuscript.

Page 9 line 4: Why are subsurface ages younger?

> Improved description in section 3:
>
> > The warming perturbation causes the AMOC and Indo-Pacific MOC to decline transiently (Fig. 1e,f, Fig. A1). The larger the forcing and implied changes in stratification, the larger the peak decline in overturning (Fig. 1e,f). The decline is likely driven by upper ocean warming, leading to increasing surface-to-deep density gradients as further modulated by salinity changes (Fig. A2). The deep ocean water mass age increases in response to the slowed overturning (Fig. 2d, 3d). As retreating sea-ice increases wind stress over these newly exposed areas, younger water masses form in the upper ocean of the Southern Ocean (Fig. 3d).

Page 9 line 9: increased stratification is not shown. Is it really increased at equilibrium or is this just a transient effect? If it is increased is this due to temperature or salin- ity?

> We add a new figure to the appendix (Figure A2c,f,i) illustrating anomalies in density for three different times (A.D. 2100, A.D. 3150, A.D. 10,000). The original paragraph referred to A.D. 3150. In new steady state conditions, the mid- and low latitudes are

> more strongly stratified. Decreases in density result from warming (Figure A2a,d,g), salinity tends to increase in the upper ocean (Figure A2b,e,h).

Fig. 3 indicates that at least in the Atlantic stratification is not increased due to temperature although export production there is decreased.

> Density is non-linear in temperature, such that the anomalies have a distinct imprint on density depending on the background temperature distribution (Figure A2).

Page 9 line 14: Why does the temperature anomaly develop there?

> This feature could well result from internal redistribution of heat as the AMOC has slowed down transiently.

Page 9 lines 18-20: Is this shown somewhere?

> We add a new figure (Figure A3).

Page 10: Part of the figure caption is missing.

> Apologies. Our version seems to be complete. We only add comments to subplots which are not self-explanatory.

Page 12 lines 4-5: what are these numbers based on?

> Based on our simulations with Bern3D across different scenarios.

Page 12 line 7: comparatively strong compared to what?

> We remove this statement. Deoxygenation in Bern3D is stronger compared to other available long-term simulations as outlined in paragraph 2 of section 5.

Page 12 lines 27-29: The discussion here is too simplistic. In the paleodata the deep oceans oxygen increased while it decreased in the thermocline. I dont see evidence provided that this is similar to the model data. It is not similar to Fig. 3a, rather the opposite, I would say.

> The original Figure 3 showed the changes at peak $O_2$ decline. The discussion on Page 12 lines 27-29, however, focused on the respective equilibrium states. We add a new Figure A3 which shows the $O_2$ anomalies for new steady state conditions for a 1.5°C warming target. The deep Pacific shows increased $O_2$ compared to PI, while most of the upper ocean shows less $O_2$ compared to PI (as a result of less solubility). As the AMOC recovers to PI values, anomalies are less pronounced in the Atlantic Ocean.

Page 12 lines 30-31: I dont agree with the statement Proxies of past ocean oxygenation and ventilation reveal similar structural changes and mecha- nisms. Increased nutrient utilization e.g. from iron fertilization also most likely played a role in glacial-interglacial changes (e.g. Schmittner and Somes, 2016, Paleoceanogr., doi:10.1002/2015PA002905).

Page 12 line 31: It is not clear if the overturning increased. Changes in overturning strength remain controversial (e.g. Kurahashi-Nakamura et al., 2016, Paleoceanogr. doi:10.1002/2016PA003001).

> We thank the reviewer for sharing his insight and accordingly remove the paleo discussion from the manuscript.

 In this study, we assess the effectiveness of the Paris climate targets in reducing hazards of decreasing oceanic oxygen, ocean warming and marine export productivity as simulated by the Bern3D  Earth system model of intermediate complexity. To this end, we prescribe in the model four different scenarios where anthropogenic GHG forcing is stabilized by 2300 AD either under stringent mitigation limiting equilibrium global

surface air warming to 1.5 or 2°C above preindustrial or following business-as-usual 21th century emissions. Simulations are run to year AD 10,000 by which time the ocean has reached new steady state conditions.  This allows us assess reversibility and the full amplitude of changes, which are larger than the near-term changes at the end of the 21st century. We summarize the outcomes developing global metrics which quantify avoided marine hazards per avoided global warming.

In section 2, we briefly describe the Bern3D model and the experimental setup. Four different radiative forcing stabilization scenarios to meet four temperature targets (1.5, 1.9, 3.3 and 9.2°C above preindustrial)  are considered. The observation-constrained 100-member ensembles used to explore parameter uncertainties for each scenario is introduced. In section 3, physical changes, including changes in overturning, water mass age, sea ice, temperature, salinity and density as well as biogeochemical changes, including changes in global oxygen inventory, the extent of oxygen minimum zones, and productivity are presented. The compound effects of warming and oxygen changes are assessed in the form of a metabolic index  (*Deutsch et al.*, 2015). Underlying physical and biogeochemical processes and mechanisms are discussed. Following earlier studies, we attribute the contributions of $O_2$ changes from changes in solubility, and the interplay of ocean biology and ventilation by carrying four explicit $O_2$ tracers and an ideal age tracer. The graphical illustration of spatial changes is focused on the 1.5 °C warming target  mentioned by the Paris Agreement, at the point of peak $O_2$ decline. Additional supporting figures are given in the appendix. In section 4, the relationship between change in global mean surface air temperature (~~SAT), atmospheric CO₂, oceanic pH, sea level and the Atlantic Meridional Overturning Circulation (AMOC)(*Plattner et al.*, 2008; *Eby et al.*, 2009; *Zickfeld et* Fewer long-term model simulations have focused on oceanic oxygen (*Yamamoto et al.*, 2015; *Matear and Hirst*, 2003; *Schmittner et al.*, 20 We show that the oceanic oxygen equilibration timescale is considerably longer than its thermal equilibration timescale and that oceanic oxygen changes are dominated by changes in Atlantic and Indo-Pacific overturning, predictive variables to be considered in future multi-millennial projections with General Circulation Models (GCMs). We also highlight that 
[revised manuscript text omitted]
. 2j). O$_2$ loss due to warming adds to transient decreases in O$_2$ utilization and diminishes the recovery level. As such, 1.5 to 3.3 °C warming targets reach similar equilibrium levels for different reasons. The degree of increased overturning from additional warming and resulting oxygenation in relation to O$_2$ loss due to higher temperatures cancel out. The 9.2 °C warming target reaches a lower equilibrium inventory compared to preindustrial due to high O$_2$ loss from solubility (-44.1 Pmol).~~

The decline and recovery pattern in oxygen changes is dominated by changes in overturning. Both Atlantic and Indo-Pacific overturning are projected to slow down and recover. Decline and recovery level as well as the decline and recovery rate vary with forcing level. Generally, Bern3D projects larger slowdown for higher forcing (Fig. 2c,d). The recovery level with forcing differs among AMOC and Indo-Pacific MOC. Higher forcing levels tend to lead to lower recovery for the AMOC and higher recovery levels for the Indo-Pacific MOC. This has direct consequences for the projected global water mass age and by that for oceanic oxygen. The higher the forcing, the higher the transient increase in water mass age. The decrease in global water mass age, which is larger for higher forcing, is dominated by increased Indo-Pacific MOC.

[Figure]

emporal evolution of critical variables relative to 1870-1899 for the model ensembles aiming at 1.5, 1.9, 3.3 and 9.2 °C warming targets. Lines mark the median and shading marks the 90 % range of the ensemble. c) Atlantic meridional overturning is the maximum of the Atlantic and d) Indo-Pacific meridional overturning is the minimum of the Indo-Pacific meridional overturning streamfunction below 400 m depth. e) $O_{2sol}$ is the explicitly traced solubility component of oceanic oxygen. i) Oxygen-poor waters are taken as the volume of water with $O_2 < 50$ mmol m$^{-3}$.

**5  Spatial changes in physical and biological variables for a 1.5 °C warming target**

We now address spatial changes in critical variables for a single, representative ensemble member at its peak $O_2$ decline which occurs at year AD 3150 and amounts to 5 % (Fig. 4). The member eventually reaches a 1.5 $S_{peak}$ is the peak sensitivity of each variable per °C warming target. $O_2$ changes show strong spatial correlation with changes in water mass age (Fig. 4a,b). Despite the global decrease, higher $O_2$ concentrations are simulated in subsurface waters where ideal age is younger. In near surface waters, local changes in remineralization may contribute to oxygen changes. Below 2 km, overall lower $O_2$ concentrations are simulated compared to preindustrial. Highest decreases are simulated in bottom waters in line with older water mass age. The presented gradients at peak $O_2$ decline tend to be more pronounced for higher forcings.

Export of particulate organic matter is simulated to increase in high latitudes and decrease elsewhere (Fig. 4c). Decreases in export production result from increased stratification and a concomitant increase in nutrient limitation in low latitudes (Fig. 4d, see also *Steinacher et al.* (2009); *Battaglia and Joos* (2017)). The increases in export production in the Arctic and Southern Ocean are due to less temperature and light limitation as surface waters warm and sea ice retreats.

The temperature anomaly is strongest within the upper ocean and decreases with depth. A pronounced temperature anomaly develops in the

**5 Uncertainties in O$_2$ projections**

The pattern and magnitude of simulated global O$_2$ changes are determined by the response of the overturning circulation.

20 O$_2$ loss due to less O$_2$ solubility at higher temperatures gradually decreases oceanic O$_2$, in addition. ~~In Bern3D, strong deoxygenation in all basins is projected to peak long after the end of the 21st century, after year AD 3000 and new steady state conditions establish after AD 8000. The equilibration timescale of oceanic oxygen is therefore longer than the thermal equilibration timescale of both the atmosphere (~1000 years) and the ocean (~4000 years).~~ Only few multi-millennial simulations with GCMs currently exist. The response of the overturning circulation on long timescales differs among available model

25 simulations (including EMICs and GCMs). Uncertainties in the equilibrium climate sensitivity additionally impact projections of O$_2$ loss due to solubility. These uncertainties directly impact projections of oceanic oxygen.

Similar circulation dynamics as simluated here (Fig. 1e,f) were found by *Rugenstein et al.* (2016) based on EMIC simulations  over 10,000 years with ECBILT-CLIO, which features a dynamic, quasi-geostrophic atmosphere.

30 *Schmittner et al.* (2008), too, found similar AMOC and Indo-Pacific MOC characteristics for their EMIC UVic 2.7, which includes an atmospheric energy balance model with fixed wind fields similar to the Bern3D model, over a 2000 year simulation. *Yamamoto et al.* (2015), on the other hand, found different overturning characteristics in a  simulation with a state-of-the art Earth System Model (MIROC 3.2 for a 4xCO$_2$) over 2000 years. There AMOC slowed down with no recovery, while AABW decreased only slightly and gradually increased thereafter. Predictions of AMOC have received more attention so far, and AMOC slowdown and partial or full recovery emerges in other multi-millennial simulations (*Zickfeld et al.*, 2013; *Li et al.*, 2013; *Weaver et al.*, 2012). AMOC and Southern Ocean overturning in CMIP5  Earth System Models was

5 analyzed by *Heuzé et al.* (2015). They found AMOC and Southern Ocean overturning is positively correlated in most CMIP5 models by the end of the 21st century. Generally, preindustrial circulation states, magnitudes and timing of changes are highly model and scenario dependent such that the long-term evolution of meridional overturning is uncertain. As oxygen changes are dominated by circulation changes, this makes the oxygen prediction highly model and scenario dependent, as well. The simulated timing and strength of peak O$_2$ decrease in Bern3D is similar to what *Schmittner et al.* (2008, AD 3000, 30 %

10 for SRES A2 high emission scenario/SAT~10 °C in Uvic 2.7) found. Other comparable simulations show earlier peaks and smaller magnitudes (*Mathesius et al.* (2015, AD 2600, 16 % decrease for RCP8.5/ΔSAT~7 °C in CLIMBER-3$\alpha$), *Yamamoto et al.* (2015, after 800 model years, 10 % for 4xCO$_2$/ΔSAT~8.5 °C in MIROC 3.2).

15

20  ~~upper ocean due to less solubility (*Jaccard et al.*, 2014). The process attribution for both paleo proxies and our long term Earth System projections including 1.5 to 3.3 °C warming targets are therefore similar. For very large radiative forcing and climate change, such as realized in the RCP8.5 scenario, projected ocean oxygen, however, remains below current concentrations even after reaching a new equilibrium. In Bern3D, changes in ventilation generally outweigh changes in remineralization fluxes as actual driver of oxygen changes. For example, oxygen concentrations decrease in the deep ocean of the low latitude~~
25  ~~and North Pacific despite lower remineralization fluxes there. At intermediate depth, younger water masses and reduced remineralization fluxes contribute to higher $O_2$ concentrations. This is in contrast to the mechanisms of $O_2$ changes identified by *Praetorius et al.* (2015) for the last deglacial transition. They postulate abrupt warming triggered expansion of the North Pacific OMZ at intermediate depth through reduced oxygen solubility and increased productivity there. We note, however, that close comparisons across the different climate states and different climate evolutions remain tentative~~

[revised manuscript text omitted]
. At peak $O_2$ loss, deep-sea environments are projected to be prone to largest changes: Large $O_2$ loss and slight warming contribute to less metabolic viability. Potential metabolic benefit from increased $O_2$ concentrations, which may develop at peak $O_2$ loss in subsurface waters, are offset by increased temperatures in most places. After transient deoxygenation, the future oceanic oxygen inventory in a 1.5 to 3.3 °C warmer world may well exceed preindustrial conditions. Under new steady state conditions, increased metabolic indices develop in better ventilated waters of the Southern Ocean and deep Indo-Pacific, despite higher temperatures. Yet, under high anthropogenic emissions and forcings such as projected in the RCP8.5 scenario, the total ocean oxygen inventory and metabolic

5 viability is reduced compared to today.

[revised manuscript text omitted]

Ehlert, D., and K. Zickfeld (2017), What determines the warming commitment after cessation of CO2 emissions?, *Environmental Research Letters*, *12*(1), 015,002.

5   Frölicher, T. L., F. Joos, G.-K. Plattner, M. Steinacher, and S. C. Doney (2009), Natural variability and anthropogenic trends in oceanic oxygen in a coupled carbon cycle–climate model ensemble, *Global Biogeochemical Cycles*, *23*(1), doi:10.1029/2008GB003316, gB1003.

Garcia, H. E., R. A. Locarnini, T. P. Boyer, J. I. Antonov, O. K. Baranova, M. M. Zweng, J. R. Reagan, and D. R. Johnson (2014), World Ocean Atlas 2013, Volume 3: Dissolved Oxygen, Apparent Oxygen Utilization, and Oxygen Saturation, *NOAA Atlas NESDIS 75 75*.

Gattuso, J.-P., A. Magnan, R. Billé, W. W. L. Cheung, E. L. Howes, F. Joos, D. Allemand, L. Bopp, S. R. Cooley, C. M. Eakin, O. Hoegh-
10   Guldberg, R. P. Kelly, H.-O. Pörtner, A. D. Rogers, J. M. Baxter, D. Laffoley, D. Osborn, A. Rankovic, J. Rochette, U. R. Sumaila, S. Treyer, and C. Turley (2015), Contrasting futures for ocean and society from different anthropogenic CO2 emissions scenarios, *Science*, *349*(6243), doi:10.1126/science.aac4722.

Golledge, N. R., D. E. Kowalewski, T. R. Naish, R. H. Levy, C. J. Fogwill, and E. G. W. Gasson (2015), The multi-millennial Antarctic commitment to future sea-level rise, *Nature*, *526*(7573), 421–425, doi:10.1038/nature15706.

15   Gruber, N. (2011), Warming up, turning sour, losing breath: ocean biogeochemistry under global change, *Philos. T. R. Soc. A*, *369*(1943), 1980–1996, doi:10.1098/rsta.2011.0003.

Heuzé, C., K. J. Heywood, D. P. Stevens, and J. K. Ridley (2015), Changes in Global Ocean Bottom Properties and Volume Transports in CMIP5 Models under Climate Change Scenarios, *Journal of Climate*, *28*(8), 2917–2944, doi:10.1175/JCLI-D-14-00381.1.

Hofmann, M., and H.-J. Schellnhuber (2009), Oceanic acidification affects marine carbon pump and triggers extended marine oxygen holes,
20   *Proceedings of the National Academy of Sciences*, *106*(9), 3017–3022, doi:10.1073/pnas.0813384106.

IPCC (2013), Climate Change 2013: The Physical Science Basis. Contribution of Working Group I to the Fifth Assessment Report of the Intergovernmental Panel on Climate Change , p. 1535 pp, Cambridge Univ. Press, United Kingdom and New York, NY, USA, doi:10.1038/446727a.

Ito, T., M. J. Follows, and E. A. Boyle (2004), Is AOU a good measure of respiration in the ocean? , *Geophys. Res. Lett.*, *31*.

25    Ito, T., S. Minobe, M. C. Long, and C. Deutsch (2017), Upper ocean O2 trends: 1958–2015, *Geophysical Research Letters*, *44*(9), 4214–4223.

Jaccard, S., E. D. Galbraith, T. L. Frölicher, and N. Gruber (2014), Ocean (De)oxygenation Across the Last Deglaciation: Insights for the Future, *Oceanography*, *27*.

Jaccard, S. L., and E. D. Galbraith (2012), Large climate-driven changes of oceanic oxygen concentrations during the last deglaciation, *Nature Geosci*, *5*(2), 151–156, doi:10.1038/ngeo1352, 10.1038/ngeo1352.

30    Jaccard, S. L., E. D. Galbraith, A. Martínez-García, and R. F. Anderson (2016), Covariation of deep Southern Ocean oxygenation and atmospheric CO2 through the last ice age, *Nature*, *530*(7589), 207–210.

Joeri, R., d. E. Michel, H. Niklas, F. Taryn, F. Hanna, W. Harald, S. Roberto, S. Fu, R. Keywan, and M. Malte (2016), Paris Agreement climate proposals need a boost to keep warming well below 2 °C, *Nature*, *534*, 631.

Joos, F., and R. Spahni (2008), Rates of change in natural and anthropogenic radiative forcing over the past 20,000 years, *P. Natl. Acad. Sci.*
35    *USA*, *105*(5), 1425–1430, doi:10.1073/pnas.0707386105.

Kalnay, E., M. Kanamitsu, R. Kistler, W. Collins, D. Deaven, L. Gandin, M. Iredell, S. Saha, G. White, J. Woollen, Y. Zhu, M. Chelliah, W. Ebisuzaki, W. Higgins, J. Janowiak, K. C. Mo, C. Ropelewski, J. Wang, A. Leetmaa, R. Reynolds, R. Jenne, and D. Joseph (1996), The NCEP/NCAR 40-year reanalysis project, *B. Am. Meteorol. Soc.*, *77*(3), 437–471, doi:10.1175/1520-0477(1996)077<0437:TNYRP>2.0.CO;2.

Keeling, R. F., A. Körtzinger, and N. Gruber (2010), Ocean Deoxygenation in a Warming World, *Annual review of marine science*, *2*, 199–229.

5    Key, R. M., A. Kozyr, C. L. Sabine, K. Lee, R. Wanninkhof, J. L. Bullister, R. A. Feely, F. J. Millero, C. Mordy, and T.-H. Peng (2004), A global ocean carbon climatology: Results from Global Data Analysis Project (GLODAP), *Global Biogeochem. Cy.*, *18*(4), GB4031, doi:10.1029/2004GB002247.

Laufkötter, C., J. G. John, C. A. Stock, and J. P. Dunne (2017), Temperature and oxygen dependence of the remineralization of organic matter, *Global Biogeochemical Cycles*, doi:10.1002/2017GB005643, 2017GB005643.

10    Li, C., J.-S. von Storch, and J. Marotzke (2013), Deep-ocean heat uptake and equilibrium climate response, *Climate Dynamics*, *40*(5), 1071–1086, doi:10.1007/s00382-012-1350-z.

Lord, N. S., A. Ridgwell, M. C. Thorne, and D. J. Lunt (2016), An impulse response function for the "long tail" of excess atmospheric CO2 in an Earth system model, *Global Biogeochemical Cycles*, *30*(1), 2–17, doi:10.1002/2014GB005074, 2014GB005074.

[revised manuscript text omitted]

Plattner, G.-K., R. Knutti, F. Joos, T. F. Stocker, W. von Bloh, V. Brovkin, D. Cameron, E. Driesschaert, S. Dutkiewicz, M. Eby, N. R.

15  Edwards, T. Fichefet, J. C. Hargreaves, C. D. Jones, M. F. Loutre, H. D. Matthews, a. Mouchet, S. a. Müller, S. Nawrath, a. Price, a. Sokolov, K. M. Strassmann, and a. J. Weaver (2008), Long-Term Climate Commitments Projected with Climate–Carbon Cycle Models, *Journal of Climate*, *21*(12), 2721–2751, doi:10.1175/2007JCLI1905.1.

Pörtner, H.-O. (2010), Oxygen- and capacity-limitation of thermal tolerance: a matrix for integrating climate-related stressor effects in marine ecosystems, *Journal of Experimental Biology*, *213*(6), 881–893, doi:10.1242/jeb.037523.

20  Praetorius, S. K., A. C. Mix, M. H. Walczak, M. D. Wolhowe, J. A. Addison, and F. G. Prahl (2015), North Pacific deglacial hypoxic events linked to abrupt ocean warming, *Nature*, *527*(7578), 362–366.

Randerson, J. T., K. Lindsay, E. Munoz, W. Fu, J. K. Moore, F. M. Hoffman, N. M. Mahowald, and S. C. Doney (2015), Multi-century changes in ocean and land contributions to the climate-carbon feedback, *Global Biogeochemical Cycles*, *29*(6), 744–759, doi:10.1002/2014GB005079, 2014GB005079.

25  Ridgwell, A., and D. N. Schmidt (2010), Past constraints on the vulnerability of marine calcifiers to massive carbon dioxide release, *Nature Geosci*, *3*(3), 196–200, doi:10.1038/ngeo755, 10.1038/ngeo755.

Ritz, S. P., T. F. Stocker, and F. Joos (2011), A coupled dynamical ocean-energy balance atmosphere model for paleoclimate studies, *J. Climate*, *24*(2), 349–375, doi:10.1175/2010JCLI3351.1.

Roth, R., S. P. Ritz, and F. Joos (2014), Burial-nutrient feedbacks amplify the sensitivity of carbon dioxide to changes in organic matter
30   remineralisation, *Earth System Dynamics*, *5*(1), 321–343, doi:10.5194/esdd-5-473-2014.

Rugenstein, M. A. A., J. Sedláček, and R. Knutti (2016), Nonlinearities in patterns of long-term ocean warming, *Geophysical Research Letters*, *43*(7), 3380–3388, doi:10.1002/2016GL068041, 2016GL068041.

Sanderson, B. M., B. C. O'Neill, and C. Tebaldi (2016), What would it take to achieve the Paris temperature targets?, *Geophysical Research Letters*, *43*(13), 7133–7142, doi:10.1002/2016GL069563, 2016GL069563.

35  Schmidtko, S., L. Stramma, and M. Visbeck (2017), Decline in global oceanic oxygen content during the past five decades, *Nature*, *542*(7641), 335–339, doi:10.1038/nature21399.

Schmittner, A., A. Oschlies, H. D. Matthews, and E. D. Galbraith (2008), Future changes in climate, ocean circulation, ecosystems, and biogeochemical cycling simulated for a business-as-usual CO2 emission scenario until year 4000 AD, *Global Biogeochem. Cy.*, *22*(1), GB1013——, doi:10.1029/2007GB002953.

Shaffer, G., S. M. Olsen, and J. O. P. Pedersen (2009), Long-term ocean oxygen depletion in response to carbon dioxide emissions from fossil fuels, *Nature Geosci*, *2*(2), 105–109, doi:10.1038/ngeo420, 10.1038/ngeo420.

5   Shakun, J. D., P. U. Clark, F. He, S. A. Marcott, A. C. Mix, Z. Liu, B. Otto-Bliesner, A. Schmittner, and E. Bard (2012), Global warming preceded by increasing carbon dioxide concentrations during the last deglaciation, *Nature*, *484*(7392), 49–54, doi:10.1038/nature10915.

Steinacher, M., and F. Joos (2016), Transient Earth system responses to cumulative carbon dioxide emissions: linearities, uncertainties, and probabilities in an observation-constrained model ensemble, *Biogeosciences*, *13*(4), 1071–1103, doi:10.5194/bg-13-1071-2016.

Steinacher, M., F. Joos, L. Bopp, P. Cadule, S. C. Doney, M. Gehlen, B. Schneider, and J. Segschneider (2009), Projected 21st century
10   decrease in marine productivity : a multi-model analysis, *Biogeosciences*, pp. 7933–7981.

Steinacher, M., F. Joos, and T. F. Stocker (2013), Allowable carbon emissions lowered by multiple climate targets, *Nature*, *499*, 197–201, doi:10.1038/nature12269.

Storch, D., L. Menzel, S. Frickenhaus, and H. O. Pörtner (2014), Climate sensitivity across marine domains of life: limits to evolutionary adaptation shape species interactions, *Global Change Biology*, *20*(10), 3059–3067.

15  Sweetman, A. K., A. Thurber, C. Smith, L. Levin, C. Mora, and C. L. Wei (2017), Major impacts of climate change on deep-sea benthic ecosystems, *Elem Sci Anth*, *5:4*.

Taucher, J., L. T. Bach, U. Riebesell, and A. Oschlies (2014), The viscosity effect on marine particle flux: A climate relevant feedback mechanism, *Global Biogeochemical Cycles*, *28*(4), 415–422, 2013GB004728.

Tschumi, T., F. Joos, and P. Parekh (2008), How important are Southern Hemisphere wind changes for low glacial carbon dioxide? A model study, *Paleoceanography*, *23*, PA4208, doi:10.1029/2008PA001592.

Tschumi, T., F. Joos, M. Gehlen, and C. Heinze (2011), Deep ocean ventilation, carbon isotopes, marine sedimentation and the deglacial
710   CO2 rise, *Clim. Past*, *7*(3), 771–800, doi:10.5194/cp-7-771-2011.

UNFCCC (accessed 11. October 2017), United Nations Framework Convention on Climate Change, Adoption of The Paris Agreement, http://unfccc.int/paris_agreement/items/9485.php.

UNFCCC (accessed 6. February 2018), United Nations Framework Convention on Climate Change, https://unfccc.int/resource/docs/convkp/conveng.pdf.

Weaver, A. J., J. Sedláček, M. Eby, K. Alexander, E. Crespin, T. Fichefet, G. Philippon-Berthier, F. Joos, M. Kawamiya, K. Matsumoto, M. Steinacher, K. Tachiiri, K. Tokos, M. Yoshimori, and K. Zickfeld (2012), Stability of the Atlantic meridional overturning circulation: A model intercomparison, *Geophysical Research Letters*, *39*(20).

Weiss, R. (1974), Carbon dioxide in water and seawater: The solubility of a non-ideal gas, *Mar. Chem.*, *2*, 203–215.

Winkelmann, R., A. Levermann, A. Ridgwell, and K. Caldeira (2015), Combustion of available fossil fuel resources sufficient to eliminate the Antarctic Ice Sheet, *Science Advances*, *1*(8), doi:10.1126/sciadv.1500589.

Yamamoto, A., A. Abe-Ouchi, M. Shigemitsu, A. Oka, K. Takahashi, R. Ohgaito, and Y. Yamanaka (2015), Global deep ocean oxygenation by enhanced ventilation in the Southern Ocean under long-term global warming, *Global Biogeochemical Cycles*, *29*(10), 1801–1815, doi:10.1002/2015GB005181, 2015GB005181.

Zickfeld, K., M. Eby, A. J. Weaver, K. Alexander, E. Crespin, N. R. Edwards, A. V. Eliseev, G. Feulner, T. Fichefet, C. E. Forest, P. Friedling-stein, H. Goosse, P. B. Holden, F. Joos, M. Kawamiya, D. Kicklighter, H. Kienert, K. Matsumoto, I. I. Mokhov, E. Monier, S. M. Olsen, J. O. P. Pedersen, M. Perrette, G. Philippon-Berthier, A. Ridgwell, A. Schlosser, T. S. V. Deimling, G. Shaffer, A. Sokolov, R. Spahni, M. Steinacher, K. Tachiiri, K. S. Tokos, M. Yoshimori, N. Zeng, and F. Zhao (2013), Long-Term Climate Change Commitment and Reversibility: An EMIC Intercomparison, *Journal of Climate*, *26*(16), 5782–5809, doi:10.1175/JCLI-D-12-00584.1.

Zickfeld, K., S. Solomon, and D. M. Gilford (2017), Centuries of thermal sea-level rise due to anthropogenic emissions of short-lived greenhouse gases, *Proceedings of the National Academy of Sciences*, *114*(4), 657–662.

---

## Author Response (AR2)

Reply to 2nd reviews for the manuscript

**Hazards of decreasing marine oxygen: the near-term and millennial-scale benefits of meeting the Paris climate targets**

submitted to Earth Syst. Dynam. Discuss.

by Gianna Battaglia, Fortunat Joos

April 13, 2018

We thank Andreas Oschlies and Andreas Schmittner for reviewing our manuscript for the second time. We much appreciate the effort and time committed by the reviewers.

Please find below our response to the comments by the reviewers and suggested text additions to the manuscript. A new manuscript, and a manuscript version with track changes are attached.

**Referee #1 Andreas Oschlies**

The authors have substantially rewritten their manuscript and satisfactorily addressed many of the concerns raised in my earlier review. Overall, the paper is more focused and shows very interesting and new scientific results about long-term ocean deoxygenation. These should be published.

The link to the Paris agreement is, however, still weak and confusing. In the model, three time points are investigated: year 2100, the time of peak change o a variable, and the models equilibrium response around year 8000. The official text of the Paris agreement does not refer to a specific time, and it would be helpful to clearly state which time the authors refer to when speaking about an X degree climate target or, even specifically, the climate targets of the Paris agreement (e.g. p.2, l.24). On p.4,l.23 the authors cite text from the UNFCCC web page that, in contrast to the Paris agreement, describes temperature targets as targets for THIS CENTURY. This is an interesting deviation from the official document. In order to avoid more confusion, the authors should take the opportunity to state more clearly what times they refer to. They mostly seem to refer to equilibrium warming, which, however, would be in conflict with their above reading of the Paris agreement.

Thank you for this comment. The revised manuscript text now only refers to the official text of the Paris Agreement.

We modified the corresponding text and added a verbatim quote from Article 2a for the 2°C and 1.5°C climate targets and from Article 4 regarding the timing of the emission reduction both taken from the official Report of the Conference of the Parties on its twenty-first session, held in Paris from 30 November to 13 December 2015. The Paris Agreement is not explicitly stating whether its climate target should be seen as equilibrium or peak warming and no specific temperature trajectory towards the temperature target is provided. Article 4 refers to a 'long-term temperature goal' and provides a time frame for greenhouse gas emission reductions. We quote now the original text from Art. 2a and Art. 4 of the Agreement to be as specific as possible regarding target and time frame, while strictly avoiding any personal interpretations of the Paris Agreement.

We further scrutinized the manuscript for clarity and added text at a few places to state explicitly what time or timeframe specific text refers to.

We added text to further emphasize the importance of long-term, multi-millennial simulations to assess climate hazards in the oceans. The introduction is slightly re-arranged for reading flow.

The modified or added text in the abstract and introduction reads as follows (bracktes refer to the attached track-changed revised version of the manuscript):

(p.1, l.4) Here, we model .. for a range of equilibrium temperature targets.

(p.2, l.10) Explicit quantification of the benefits of meeting the 2°C or 1.5°C climate targets mentioned by the Paris Agreement with respect to the reversibility and avoidance of implied impacts on marine oxygen and related environmental parameters, including ocean circulation, ocean warming, metabolic viability and biological productivity on multi-millennial timescales is missing.

(p.4, l.1) A goal of the Paris Agreement is to 'hold the increase in the global average temperature to well below 2°C above pre-industrial levels and pursuing efforts to limit the temperature increase to 1.5°C above pre-industrial levels' (Article 2a, *Paris Agreement*, accessed 10. April 2018). Article 4 of the Paris Agreement further states that 'In order to achieve the long-term temperature goal set out in Article 2, Parties aim to reach global peaking of greenhouse gas emissions as soon as possible .. and to undertake rapid reductions thereafter in accordance with best available science, so as to achieve a balance between anthropogenic emissions by sources and removals by sinks of greenhouse gases in the second half of this century (Article 4, *Paris Agreement*, accessed 10. April 2018).

(p.4, l.16) We prescribe in the model four different, idealized scenarios where anthropogenic GHG forcing is stabilized by 2300 AD. The four scenarios are designed to reach an equilibrium warming of 1.5, 1.9, 3.3 and 9.2°C above preindustrial. Simulations are run to year AD 10,000 by which time the ocean has reached new steady state conditions. This allows us to assess reversibility and the full amplitude

of changes, acknowledging the long equilibration timescale of biogeochemical variables with peak hazards potentially occurring long after stabilization of radiative forcing in the atmosphere. We summarize the outcomes developing global metrics which quantify avoided marine hazards per avoided global warming on three different time horizons. The first time horizon is the end of the 21st century, the typical assessment timescale of climate change hazards. Here, changes in a variable are related to changes in SAT at year 2100. Those are contrasted to the millennial-scale perspective where peak changes in the variable in the course of the simulation and equilibrium changes at the end of the simulation are related to the corresponding equilibrium warming.

(p.5, l.4) The graphical illustration of spatial changes is focused on a 1.5°C equilibrium warming target at the point of peak $O_2$ decline.

(p.5, l.8) Often a near-linear relationship is found between the change in a variable of interest and the change in SAT as simulated across the range of scenarios and ensemble members at a distinct time. This allows us to develop new metrics to quantify avoided marine hazards per unit change in $\Delta SAT$ for different points in time.

New text is also added to the second paragraph of section 4:

(p.16,l.10) We compare and contrast changes at the end of the 21st century, the typical assessment timescale of climate change, to peak and equilibrium changes at the millennial timescale. Peak and equilibrium changes are related to the corresponding equilibrium temperature response, while changes at the end of the 21st century are related to the transiently realized warming at the end of the 21st century.

Please also refer to the attached manuscript with track changes.

individual points:

p.5, l.4 are equilibrium changes always larger ?

Sentence is adjusted for more general validity.

> This allows us to assess reversibility and the full amplitude of changes, acknowledging the long equilibration timescale of biogeochemical variables with peak hazards potentially occurring long after stabilization of radiative forcing in the atmosphere.

p.5 l.5 what is the time stamp of global warming here?

> Further explanation is given highlighting the decline-recovery phase behavior of biogeochemical variables with peak hazards potentially occurring long after stabilization of radiative forcing in the atmosphere.

p.5, l.31 It is not clear what is meant by linear. Linear in time, in the equilibrium temperature? p.5, l.32 time stamp of $\Delta$ SAT ?

> Sentence is clarified by:
>
> > Often a near-linear relationship is found between the change in a variable of interest and the change in SAT as simulated across the range of scenarios and ensemble members at a distinct time. This allows us to develop new metrics to quantify avoided marine hazards per unit change in $\Delta$SAT for different points in time.

p.10,l.23 correlates is difficult to see from Fig.2 - all curves somehow correlate with each other. Describe more clearly what you mean.

> We add the following sentence to this paragraph:
>
> > The $O_2$ consumption tracer (Fig. 2e) determines the shape of the global $O_2$ signal (Fig. 2a). Its decline and recovery phase is strongly correlated with the evolution of ideal age (Fig. 2e and Fig. 2d, see also Fig. 3b and Fig. 3d). As has been shown

> previously (e.g. *Yamamoto et al.*, 2015; *Bopp et al.*, 2017), the high correlation between changes in $O_2$ and ideal age and the absence of a direct relationship between changes in remineralization fluxes and $O_2$, indicate that circulation changes are the major contributors to changes in $O_2$.

p.12 Fig 1c,d. Mention whether these are annual mean or summer ice areas.

> Thanks for this comment. Variables refer to annual mean sea-ice areas. The text is updated at several locations and a note is added to the label of Figure 1:
>
> c) and d) are annual mean sea-ice areas in the respective hemisphere.

p.16, l.31 should probably refer to Fig.5a

> Correct. The reference was correct in the revised version of the manuscript, but wrongly assigned in the track changed version.

p.23, l.13. 320 GtC post-2017 emissions would be a larger carbon budget for the 1.5C target than implied by the last IPCC AR5, I think. I looked up the Steinacher et al. (2013) paper, but could not find this number.

> Text added in parentheses at end of sentence for clarification:
>
> (the 320 GtC are derived by adjusting the emission limit as displayed in Fig. 4 of *Steinacher et al.* (2013) for year 2000 using fossil emissions published by the Global Carbon Project).
>
> Please see *Millar et al.* (2017) for a general disucssion on compatible emission estimates.

**Referee #2 Andreas Schmittner**

The authors have responded to my previous comments and changed the manuscript appropriately. I think it is ready for publication. I only have the following few minor additional comments.

Page 8, lines 13-14: I could imagine reduced convection as an alternative explanation for the younger upper ocean because it would decrease the entrainment of old deep waters.

> We add a sentence acknowledging this point:
>
> > Reduced convection may contribute to a younger upper ocean through decresed entrainment of old deep water (not quantified within the scope of this paper).

Page 16 lines 12-13: I think panels (i) in Fig. (2), which show denitrification are wrong in this context. In line 13 I think it should refer to panel (g), which shows export production.

> Thanks, we updated the references to Fig. 2.

Page 19, line 18: The resulting impact on the nitrogen cycle has been quantified in model simulations before and thus it may not be completely unclear. E.g. Schmittner et al. (2008) have simulated a decrease in the fixed nitrogen inventory due to increases in denitrification.

> We remove the mentioned sentence from this paragraph, as it may be perceived as a too strong statement. This concluding sentence referred to the different aspects mentioned in the entire paragraph, not just the impact of denitrification on the N-cycle.

**References**

[revised manuscript text omitted]